

# Higgs associated production with a vector decaying to two fermions in the geoSMEFT

Tyler Corbett[1][*] and Adam Martin[2][†]

**1** Faculty of Physics, University of Vienna, Boltzmanngasse 5, A-1090 Wien, Austria
**2** Department of Physics, University of Notre Dame, Notre Dame, IN, 46556, USA

[*] corbett.t.s@gmail.com , [†] amarti41@nd.edu

## Abstract

We present the inclusive calculations of a Higgs boson produced in association with massive vector bosons in the Standard Model Effective Field Theory (SMEFT) to order $1/\Lambda^4$ for the 13 TeV LHC. The calculations include the decay of the vector boson into massless constituents and are done using the geometric formulation of the SMEFT supplemented by the relevant dimension eight operators not included in the geoSMEFT. We include some discussion of distributions to motivate how detailed collider and experimental searches for SMEFT signals could be improved.

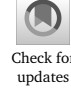

# 1   Introduction

The Standard Model (SM) very accurately describes phenomena up to the electroweak scale. However, given the failings of the SM to describe certain phenomena, such as neutrino masses, dark matter, and the baryon asymmetry of the universe, we know physics beyond the SM exists. Having concluded run two, and with the start of run three, the LHC has yet to directly discover physics beyond the SM.

Given the lack of evidence of new physics from the LHC, we are interested in exploring evidence for physics beyond the SM that the LHC cannot directly produce. A natural approach to such a study is the use of Effective Field Theories (EFTs). One of the most studied EFT approaches to physics beyond the SM is the Standard Model EFT (SMEFT). In this approach a tower of operators is added to the SM in order to quantify the effects of new physics:

$$\mathcal{L}_{\text{SMEFT}} = \mathcal{L}_{\text{SM}} + \sum_{i,j} \frac{1}{\Lambda^i} Q_j^{(4+i)} \,. \tag{1}$$

Here $i$ represents the sum over all operators of dimension $4 + i > 4$ while $j$ indicates a sum over all possible operators at a given dimension. In the SMEFT one also assumes that the Higgs boson is embedded into an $SU(2)_L$ doublet. Operators of odd dimension necessarily include lepton and/or baryon number violation. As such the leading order for most processes considered for LHC physics is dimension six, or $1/\Lambda^2$. For examples of recent global fits to Higgs, electroweak, and/or top physics at order $1/\Lambda^2$ see e.g. [1–3].

More recently a great deal of interest has been generated in studying the next order in the SMEFT expansion, $1/\Lambda^4$ [4–16]. Motivations for the study of these NLO terms in the SMEFT expansion include:

1. The "inverse problem," or that there are a large number of UV completions of the SM which imprint on the same subset of operators at dimension six. This degeneracy is largely broken at dimension eight [17], and therefore leads to the hope that if indirect evidence of physics beyond the SM is discovered we can better understand the nature of the new physics.

2. Truncation error in the SMEFT is not well understood. The most common approach is to consider squares of dimension six operators as an estimate of the error [18]. However this is not consistent with the power counting of the SMEFT: inclusion of dimension-six-squared contributions necessitates the inclusion of dimension-eight operators as both occur at order $1/\Lambda^4$. An alternative approach to truncation error uses the dimension-eight terms as nuisance parameters in order to study the impact of neglecting them and thereby infer the truncation error [19, 20].

3. There are known instances where the dimension-eight terms have significant impacts on the infrared physics that would be missed by only including dimension-six results

[14, 21–24]. This should, for completeness, be contrasted with cases where dimension-eight has little phenomenological impact [12].

With this in mind this article seeks to continue to push phenomenological studies of LHC processes to order $1/\Lambda^4$. In the Higgs sector gluon fusion production to order $1/\Lambda^4$ at tree level is already known,[1] as are the decays of the Higgs boson to two particles which includes the largest and most accurately measured decay products, $h \to \bar{b}b$ and $h \to \gamma\gamma$. Higgs boson associated production with a vector is particularly interesting as the geoSMEFT approach allows us to simply derive many of the properties of the process which primarily depends on three-point vertices for which the geoSMEFT is particularly well suited.

In this article we calculate the Higgs boson production in association with a massive vector boson, $pp \to VH$, where $V = W^{\pm}/Z$. We utilize the geoSMEFT methodology which greatly simplifies a large part of the calculations including the choice of input parameter schemes [23, 26]. In order to include the remaining terms at order $1/\Lambda^4$ not included in the geoSMEFT we follow the methods developed in [10]. To produce results closer to what is actually measured in LHC experiments, we include the decay of the vector boson into massless leptons in our calculation (along with all SMEFT effects contained in the decay), $pp \to V(\bar{f}f)H$. Using this $2 \to 3$ process, we can impose realistic triggering and analysis cuts. The relevant diagrams for this process are shown in Fig. 1. Notice Fig. 1(b) and (c) are not present in the SM. While we include the decay of the vector boson, we will neglect contributions from the last diagram, Fig. 1c, as we assume experimental searches will have a cut on $m_{\ell\ell} \sim m_{W/Z}$. This also removes the potential five-point vertex arising at dimension-eight which is not shown in Fig. 1.

The article is organized as follows: In Sec. 2 we introduce the necessary calculational ingredients, namely the geoSMEFT, dimension-eight operators not included in the geoSMEFT, and the concept of expanding the propagator to a given order in the SMEFT expansion. Then in Sec. 3 we reexpress these in terms of an effective Lagrangian which is used for the actual calculations. Section 4 contains the main results of this article, the parameterization of Higgs boson production in association with a vector boson to order $1/\Lambda^4$ in the power counting. We conclude in Sec. 5, while leaving certain relevant details of the calculations in the Appendices in order to maintain a more direct discussion in the main body of the text. Our results are all leading order in SM couplings. The effect of terms that are leading order in the SMEFT expansion ($\mathcal{O}(1/\Lambda^2)$) but higher order in SM couplings would be interesting to explore but are beyond the scope of this work.

The ancillary files include a Mathematica notebook which puts together all of the results and identities of this work in order to generate an expression for the inclusive cross sections for $ZH$ and $W^{\pm}H$ production, as well as the case of $ZH$ with partonic center of mass energy greater than 500 GeV. These results depend on a multitude of Wilson coefficients and so are poorly suited for inclusion in the text.

## 2 The geoSMEFT and dimension-eight operator bases

The geoSMEFT is a tool which reorganizes part of the SMEFT in order to fully classify all operators to all orders in the SMEFT which affect two- and three-point functions. This is achieved by considering operators involving up to three fields ($F_i$) and their derivatives, and rewriting them as a field-space connection which is a function of the SM scalar boson, $M(\phi)$ below (where $\phi$ is the scalar), multiplied by the $F_i$ considered:

$$
\begin{aligned}
F_1, F_2 &\to M_{12}(\phi)F_1F_2\,, \\
F_1, F_2, F_3 &\to M_{123}(\phi)F_1F_2F_3\,.
\end{aligned}
\tag{2}
$$

---

[1]As well as recently the contributions at one-loop from amplitudes with two insertions of dimension-six operators in [25].

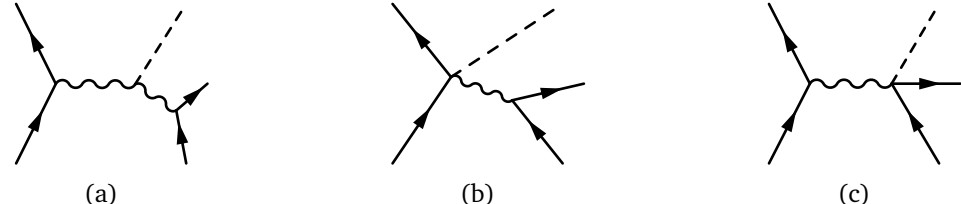

(a)            (b)            (c)

Figure 1: Diagrams contributing to Higgs associated production with a vector boson in the SMEFT. Figure (1a) includes the SM contributions while Figs. (1b) and (1c) are only generated in the SMEFT. We make the assumption that experimental searches place a $m_Z$ or $m_W$ cut on the invariant mass of the final state leptons which effectively removes the contributions from Fig. (1c) as well as potential contributions from other diagrams generated by e.g. four-fermion operators.

That the geoSMEFT classifies all three-point functions is a basis dependent statement, and therefore the geoSMEFT also defines a basis of operators in the SMEFT. The geoSMEFT is consistent with the Warsaw basis (dimension-six) [27], but requires we revisit the choice of the basis of operators at dimension-eight. This is because the two standard bases at dimension-eight as defined in [28,29] include operators which shift the three-point functions and are not in the geoSMEFT.

The "geoSMEFT" as used in this article and the literature corresponds to the choice of fully classifying all two- and three-point functions. This choice greatly simplifies calculations involving resonant physics. One could, however, apply the geometric methodology to any basis of SMEFT operators by defining a different minimal set of field-space connections.

In Section 2.1 below we discuss the operators in the geoSMEFT framework relevant to our discussion of the associated production of a Higgs boson with a vector boson, then in Section 2.2 we discuss the change of basis necessary to make the above two dimension-eight operator bases consistent with the geoSMEFT in order to obtain results for the process to order $1/\Lambda^4$.

## 2.1 The geoSMEFT and associated production

Defining $\phi^I$ as the complex scalar doublet of the SM rewritten as a four-component real field, and $W_\mu^A$ is a four component real vector field containing the three $SU(2)_L$ and the $U(1)_Y$ gauge fields, we can write the relevant terms from the geoSMEFT [26,30] (i.e. those which generate vertices contributing to the diagrams in Fig. 1):

$$\mathcal{L}_{\text{geoSMEFT}} = \frac{1}{2} h_{IJ}(D_\mu \phi)^I (D^\mu \phi)^J + \frac{1}{4} g_{AB} W_{\mu\nu}^A W^{B,\mu\nu} + \kappa_{IJ}^A (D_\mu \phi)^I (D_\nu \phi)^J W^{A,\mu\nu} \tag{3}$$
$$+ L_{J,A}^{\psi,pr}(D_\mu \phi)^J (\bar{\psi}_p \gamma_\mu \tilde{\sigma}^A \psi_r) + L_J^{ud,pr}(D_\mu \phi)^J (\bar{u}_p \gamma^\mu d_r)$$
$$+ \left( d_A^{pr} \bar{\psi}_p \sigma^{\mu\nu} \psi_r W_{\mu\nu}^A + h.c. \right),$$

where $p$ and $r$ are flavor indices, and we have defined:

$$\tilde{\sigma}^A = \delta^{A,4} + (1 - \delta^{A,4})\sigma^A. \tag{4}$$

The field-space connections $h_{IJ}$ and $g_{AB}$ shift all Higgs and vector boson interactions respectively, while $\kappa_{IJ}^A$, $L_{J,A}^\psi$, $L_J^{ud}$, and $d_A$ generate three- and four-point functions contributing to the diagrams of Fig. 1.[2] For simplicity we will assume a $U(3)^5$ flavor symmetry, this removes the dipole operators of the last line of Eq. 3.

---

[2]Recall from Eq. 2 and the discussion above, these field-space connections are all functions of the real scalar field $\phi^I$.

These field-space connections can be written as matrices in the $SU(2)_L$ space, and can be written to all orders as:

$$h_{IJ} = \left[1 + \phi^2 c_{H\Box}^{(6)} + \sum_{n=0}^{\infty}\left(\frac{\phi^2}{2}\right)^{n+2}\left(c_{HD}^{(8+2n)} - c_{HD,2}^{(8+2n)}\right)\right]\delta_{IJ}$$
$$+ \frac{\Gamma_{A,J}^I \phi_K \Gamma_{A,L}^K \phi^L}{2}\left(\frac{c_{HD}^{(6)}}{2} + \sum_{n=0}^{\infty}\left(\frac{\phi^2}{2}\right)^{n+1} c_{HD,2}^{(8+2n)}\right), \tag{5}$$

$$g_{AB} = \left[1 - 4\sum_{n=0}^{\infty}\left(c_{HW}^{(6+2n)}(1-\delta_{A4}) + c_{HB}^{(6+2n)}\delta_{A4}\right)\left(\frac{\phi^2}{2}\right)^{n+1}\right]\delta_{AB}$$
$$- \sum_{n=0}^{\infty}c_{HW,2}^{(8+2n)}\left(\frac{\phi^2}{2}\right)^n\left(\phi_I\Gamma_{A,J}^I\phi^J\right)\left(\phi_L\Gamma_{B,K}^L\phi^K\right)(1-\delta_{A4})(1-\delta_{B4}) \tag{6}$$
$$+ \left[\sum_{n=0}^{\infty}c_{HWB}^{(6+2n)}\left(\frac{\phi^2}{2}\right)^n\right]\left[\left(\phi_I\Gamma_{A,J}^I\phi^J\right)(1-\delta_{A4})\delta_{B4} + (A\leftrightarrow B)\right],$$

$$k_{IJ}^A = -\frac{1}{2}\gamma_{4J}^I\delta_{A4}\sum_{n=0}^{\infty}c_{HDHB}^{(8+2n)}\left(\frac{\phi^2}{2}\right)^{n+1} - \frac{1}{2}\gamma_{AJ}^I(1-\delta_{A4})\sum_{n=0}^{\infty}c_{HDHW}^{(8+2n)}\left(\frac{\phi^2}{2}\right)^{n+1}$$
$$- \frac{1}{8}(1-\delta_{A4})\left[\phi_K\Gamma_{AL}^K\phi^L\right]\left[\phi_M\Gamma_{BN}^M\phi^N\right]\gamma_{BJ}^I\sum_{n=0}^{\infty}c_{HDHW,3}^{(8+2n)}\left(\frac{\phi^2}{2}\right)^n \tag{7}$$
$$+ \frac{1}{4}\epsilon_{ABC}\left[\phi_K\Gamma_{BL}^K\phi^L\right]\gamma_{CJ}^I\sum_{n=0}^{\infty}c_{HDHW,2}^{(8+2n)}\left(\frac{\phi^2}{2}\right)^n,$$

$$L_{JA}^{\psi pr} = -\left(\phi\gamma_4\right)_J\delta_{A4}\sum_{n=0}^{\infty}c_{H\psi,pr}^{1,(6+2n)}\left(\frac{\phi^2}{2}\right)^n - \left(\phi\gamma_A\right)_J(1-\delta_{A4})\sum_{n=0}^{\infty}c_{H\psi,pr}^{3,(6+2n)}\left(\frac{\phi^2}{2}\right)^n$$
$$+ \frac{1}{2}\left(\phi\gamma_4\right)_J(1-\delta_{A4})\left(\phi_K\Gamma_{AL}^K\phi^L\right)\sum_{n=0}^{\infty}c_{H\psi,pr}^{2,(8+2n)}\left(\frac{\phi^2}{2}\right)^n \tag{8}$$
$$+ \frac{\epsilon_{BC}^A}{2}\left(\phi\gamma_B\right)_J\left(\phi_K\Gamma_{CL}^K\phi^L\right)\sum_{n=0}^{\infty}c_{H\psi,pr}^{\epsilon,(8+2n)}\left(\frac{\phi^2}{2}\right)^n. \tag{9}$$

$$d_A^{\psi,pr} = \sum_{n=0}^{\infty}\left(\frac{\phi^2}{2}\right)^n\left[\delta_{A4}c_{\psi B,pr}^{(6+2n)} + \sigma_A(1-\delta_{A4})c_{\psi W,pr}^{(6+2n)} - \left[\phi^K\Gamma_{AL}^K\phi^L\right](1-\delta_{A4})c_{\psi W,2,pr}^{(8+2n)}\right]\overset{(\sim)}{H}. \tag{10}$$

In the above, $\Gamma_{AJ}^I$ and $\gamma_{AJ}^I$ are different ways of rewriting the generators of $SU(2)_L \times U(1)_Y$ in the basis of the four-component $\phi$ and $W$. They can be found in [26]. The $c_i^{(d)}$ are the Wilson coefficients of the SMEFT for an operator with label $i$ of dimension $d$. The corresponding operator forms in the usual SMEFT formulation can also be found in Section 3 of [26].

## 2.2 A dimension-eight basis consistent with the geoSMEFT

As was mentioned above, the geoSMEFT classifies all operators which affect two- and three-point functions in the SMEFT. While such a classification includes some subset of operators affecting four-and higher point functions, in order to consistently calculate Higgs associated production to order $1/\Lambda^4$ we must include all dimension-eight operator contributions. That is, we must include operators at dimension-eight not included in the geoSMEFT. We can achieve this by considering the bases as written in [28, 29] and proceeding as follows:

1. Identify all operators affecting two- and three-point functions which are not included in the geoSMEFT.



2. Removing these operators from the dimension-eight operator basis via integration by parts and the equations of motion. That is, the operators' effects are moved to other operators consistent with the geoSMEFT.

3. Including the new operator forms introduced by the above step.

This procedure was first implemented in [10]. We can take the operators of the form $H^4D^4$ from [29] as an example. These operators are not consistent with the geoSMEFT basis. Taking the operator,

$$Q^{(1)}_{H^2H^{\dagger 2}D^4} = (H^\dagger H)\Box^2(H^\dagger H), \tag{11}$$

we see that after spontaneous symmetry breaking this operator affects both 2- and 3-point functions as $H \sim \{0, v + h\}^T$, and the terms from the $(H^\dagger H)$ on the left allows for terms like $v^2\Box^2 h^2$. This operator can be exchanged by integration by parts and the equations of motion for those of [28]:

$$Q^{(1)}_{H^4} = (D_\mu H)^\dagger(D_\nu H)(D^\nu H)^\dagger(D^\mu H), \tag{12}$$

$$Q^{(2)}_{H^4} = (D_\mu H)^\dagger(D_\nu H)(D^\mu H)^\dagger(D^\nu H), \tag{13}$$

$$Q^{(3)}_{H^4} = (D_\mu H)^\dagger(D^\mu H)(D^\nu H)^\dagger(D_\nu H). \tag{14}$$

These new operators do not generate two- or three-point functions, and are thus consistent with the geoSMEFT basis. That is, since derivatives act on all Higgs doublets, terms where a field is exchanged for the constant $v$ will vanish identically. Unrelated to the geoSMEFT discussion, it is also preferable to remove the operator of Eq. 11 as it requires the introduction of Lee-Wick ghosts [31].

Following the steps above we systematically go through the operators of references [28,29] and identify the following operator forms which are not consistent with the geoSMEFT basis. Below, we adopt the usual convention where $X$ stands for a field strength tensor, $D$ for the covariant derivative (acting on any of the field content), $\psi$ for any of the fermionic fields, and $H$ for the Higgs doublet. We find the following classes of operators to be inconsistent with the geoSMEFT:

1. Class $\psi^2 H^2 D^3$ – in both [28] and [29].

2. Class $H^4 D^4$ – is consistent in [28], but is not in [29] as mentioned in the example of Eq. 11.

All other operators in the other two bases were found to be consistent with the geoSMEFT basis. The operator classes $\psi^2 H^2 X D$, $\psi^2 H^3 D^2$, and $\psi^2 H X D^2$ in particular could have presented problems, but in both dimension-eight bases the derivatives were written acting solely on the Higgs doublets so they conform with the geoSMEFT basis.

In order to make the operators of classes $D^3\psi^2 H^2$ consistent with the geoSMEFT it is useful to use the tools developed in [26] and [29]. For illustration, the class $D^3\psi^2 H^2$ is worked out in more detail in Appendix D.

Tables 1 and 2 contain the full set of dimension-eight operators relevant to Higgs associated production and not contained in the geoSMEFT operators of Eq. 3.[3] They fall under the classes $\psi^2 X H^2 D$ and $\psi^2 H^2 D^3$. Here we only allow $\psi = \{q, u, d\}$ as we are assuming $m_{\ell\ell} \sim m_{W/Z}$ which effectively removes the contributions from Fig. 1c and therefore the contribution of leptonic operators. With all operators contributing we may now consider all effective vertices contributing to the process.

---

[3]Subject, again, to our assumption of a $U(3)^5$ flavor symmetry.

Table 1: Operators of class $\psi^2 X H^2 D$ contributing to Higgs associated production up to $\mathcal{O}(1/\Lambda^4)$. The last column indicates which chiral quarks contribute. Note our naming convention differs from that of [28] and our convention for $\overleftrightarrow{D}$ differs by a factor of $i$.

| | Operator | relevant $\psi$ |
|---|---|---|
| $Q_{\psi^2 B H^2 D}^{(1)}$ | $(\bar{\psi}_p \gamma^\nu \psi_r) D^\mu (H^\dagger H) B_{\mu\nu}$ | $\psi = \{q, u, d\}$ |
| $Q_{\psi^2 B H^2 D}^{(2)}$ | $i(\bar{\psi}_p \gamma^\nu \psi_r)(H^\dagger \overleftrightarrow{D}^\mu H) B_{\mu\nu}$ | $\psi = \{q, u, d\}$ |
| $Q_{\psi^2 B H^2 D}^{(3)}$ | $(\bar{\psi}_p \gamma^\nu \sigma^I \psi_r) D^\mu (H^\dagger \sigma^I H) B_{\mu\nu}$ | $\psi = \{q\}$ |
| $Q_{\psi^2 B H^2 D}^{(4)}$ | $i(\bar{\psi}_p \gamma^\nu \sigma^I \psi_r)(H^\dagger \overleftrightarrow{D}^{I\mu} H) B_{\mu\nu}$ | $\psi = \{q\}$ |
| $Q_{\psi^2 W H^2 D}^{(1)}$ | $(\bar{\psi}_p \gamma^\nu \psi_r) D^\mu (H^\dagger \sigma^I H) W_{\mu\nu}^I$ | $\psi = \{q, u, d\}$ |
| $Q_{\psi^2 W H^2 D}^{(2)}$ | $i(\bar{\psi}_p \gamma^\nu \psi_r)(H^\dagger \overleftrightarrow{D}^{I\mu} H) W_{\mu\nu}^I$ | $\psi = \{q, u, d\}$ |
| $Q_{\psi^2 W H^2 D}^{(3)}$ | $(\bar{\psi}_p \gamma^\nu \sigma^I \psi_r) D^\mu (H^\dagger H) W_{\mu\nu}^I$ | $\psi = \{q\}$ |
| $Q_{\psi^2 W H^2 D}^{(4)}$ | $i(\bar{\psi}_p \gamma^\nu \sigma^I \psi_r)(H^\dagger \overleftrightarrow{D}^\mu H) W_{\mu\nu}^I$ | $\psi = \{q\}$ |
| $Q_{\psi^2 W H^2 D}^{(5)}$ | $\epsilon_{IJK}(\bar{\psi}_p \gamma^\nu \sigma^I \psi_r) D^\mu (H^\dagger \sigma^J H) W_{\mu\nu}^K$ | $\psi = \{q\}$ |
| $Q_{\psi^2 W H^2 D}^{(6)}$ | $i\epsilon_{IJK}(\bar{\psi}_p \gamma^\nu \sigma^I \psi_r)(H^\dagger \overleftrightarrow{D}^{J\mu} H) W_{\mu\nu}^K$ | $\psi = \{q\}$ |

## 2.3 Propagators and the SMEFT expansion

In addition to effective vertices arising from the SMEFT, we also need to take into account that the SMEFT shifts both the gauge boson masses and widths. As such, we expand the propagators for the gauge bosons (in the unitary gauge) order by order in the SMEFT power counting. This approach was first adopted to order $1/\Lambda^2$ in [32], and has since been introduced to SMEFTsim (again to order $1/\Lambda^2$) [33]. Denoting the shifts in the mass and width as $\delta m_V$ and $\delta \Gamma_V$ we find:

$$
\frac{ig^{\mu\nu}}{p^2 - \bar{m}_V^2 + i\bar{\Gamma}_V \bar{m}_V} \to \frac{ig^{\mu\nu}}{p^2 - m_V^2 + i\Gamma_V m_V} \left[ \frac{(\Gamma_V + 2im_V)\delta m_V + m_V \delta \Gamma_V}{p^2 - m_V^2 + i\Gamma_V m_V} \right.
$$
$$
\left. + \frac{m_V^2 (\delta \Gamma_V)^2 + (\Gamma_V^2 + 3i\Gamma_V m_V - 3m_V^2 - p^2)(\delta m_V)^2 + (\Gamma_V m_V + i[p^2 + 3m_V^2])\delta \Gamma_V \delta m_V}{p^2 - m_V^2 + i\Gamma_V m_V} \right].
$$
(15)

Where it should be understood that $\delta m_V$ and $\delta \Gamma_V$ include shifts up to order $1/\Lambda^4$ and that, for this work, the amplitude squared needs to be truncated at order $1/\Lambda^4$.

In this work we consider two popular choices of input parameters: the $\{\hat{\alpha}, \hat{m}_Z, \hat{G}_F\}$ or "$\alpha$" scheme, and $\{\hat{m}_W, \hat{m}_Z, \hat{G}_F\}$ or "$m_W$" scheme. In the case of the $\hat{m}_W$ scheme, as the mass of the $W$ is an input parameter, it receives no shift $\delta m_W$ and so we do not need to include this term. In both schemes the $Z$ mass is an input and so the shift $\delta m_Z$ is never necessary. Both schemes have been worked out to arbitrary order in the SMEFT power counting in [23]. We have included the mapping from Lagrangian parameters to input parameters in Appendix A as well as other necessary ingredients to convert our geoSMEFT conventions to Wilson coefficients.

## 3 The effective Lagrangian

Next we derive the full set of effective vertices in the form of an effective Lagrangian. This is split between the effective Lagrangians for the neutral currents and the charged currents.

Table 2: Operators of class $\psi^2 H^2 D^3$ contributing to Higgs associated production up to $\mathcal{O}(1/\Lambda^4)$. The last column indicates which chiral quarks contribute. The subscript $(\mu, \nu)$ indicates the the derivatives are symmetrized in the Lorentz indices. As discussed in Sec. 2.2, the operators in this class differ from those of [28, 29] as we are using a geoSMEFT compliant basis.

| | Operator | relevant $\psi$ |
|---|---|---|
| $Q_{\psi^2 H^2 D^3}^{(1)}$ | $i(\bar{\psi}_p \gamma^\mu \psi_r)\left[(D_\nu H)^\dagger(D^2_{(\mu,\nu)}H) - (D^2_{(\mu,\nu)}H)^\dagger(D_\nu H)\right]$ | $\psi = \{q, u, d\}$ |
| $Q_{\psi^2 H^2 D^3}^{(2)}$ | $i(\bar{\psi}_p \gamma^\mu \overleftrightarrow{D}_\nu \psi_r)\left[(D_\mu H)^\dagger(D_\nu H) + (D_\nu H)^\dagger(D_\mu H)\right]$ | $\psi = \{q, u, d\}$ |
| $Q_{\psi^2 H^2 D^3}^{(3)}$ | $i(\bar{\psi}_p \gamma^\mu \sigma^I \psi_r)\left[(D_\nu H)^\dagger \tau^I(D^2_{(\mu,\nu)}H) - (D^2_{(\mu,\nu)}H)^\dagger \sigma^I(D_\nu H)\right]$ | $\psi = \{q\}$ |
| $Q_{\psi^2 H^2 D^3}^{(4)}$ | $i(\bar{\psi}_p \gamma^\mu \sigma^I \overleftrightarrow{D}_\nu \psi_r)\left[(D_\mu H)^\dagger \tau^I(D_\nu H) + (D_\nu H)^\dagger \tau^I(D_\mu H)\right]$ | $\psi = \{q\}$ |

Further we will do this separately for the geoSMEFT and operators of Sec. 2.2 within each subsection below.

## 3.1 $\mathcal{L}_{\text{eff}}$ for neutral currents

Beginning from the the geoSMEFT, as laid out in Eq. 3 and Eqs. 5–9. Using the shorthand $\psi$ for any of the four-component SM fermionic fields, the resulting effective Lagrangian is given by:

$$
\begin{aligned}
\mathcal{L}_{\text{eff}}^{\text{NC},1} &= \frac{1}{2}c_{HZZ}^{(1)}hZ_\mu Z^\mu + \frac{c_{HZZ}^{(2)}}{4}hZ_{\mu\nu}Z^{\mu\nu} + c_{HZZ}^{(3)}(\partial_\mu h)Z^{\mu\nu}Z_\nu \\
&+ \frac{1}{2}c_{HAZ}^{(2)}hA_{\mu\nu}Z^{\mu\nu} + c_{HAZ}^{(3)}(\partial_\mu h)A^{\mu\nu}Z_\nu \\
&- \bar{e}\bar{\psi}_p \gamma^\mu \psi_p A_\mu + \left(hg_{\pm,pr}^{(1),hZ\psi} + g_{\pm,pr}^{Z\psi}\right)\bar{\psi}_p \gamma^\mu P_\pm \psi_r Z_\mu .
\end{aligned}
\tag{16}
$$

The first two lines correspond to purely bosonic couplings which are completely captured by the geoSMEFT. Therefore, they only generate diagram (a) of Fig. 1. The last three lines generate $Z\bar{\psi}\psi$ and $Zh\bar{\psi}\psi$ vertices and therefore can contribute to both diagrams (a) and (b). The above effective Lagrangian allows for flavor changing neutral currents, consistent with the SMEFT, however we take all neutral current couplings to be flavor diagonal. In order to keep the main text from becoming encumbered with definitions and long expressions ill-suited for print, the dependence of the $c_i^j$ on geometrically defined quantities and the Wilson coefficients can be found in App. A.

Next we consider the operators of Tables 1 and 2. The operators do not generate three-point vertices (see Sec. 2.2) and therefore we only need to consider operators generating the effective vertex of Fig. 1(b). Our assumption that $m_{ll} \sim m_V$ removes contributions from Fig. 1(c) as well as contributions from the four-point vertices involving a photon. We find the effective Lagrangian:

$$
\begin{aligned}
\mathcal{L} &= g_\pm^{(2),hZ\psi}(\partial_\nu h)Z_\mu(\partial_\nu \bar{\psi})\gamma^\mu P_\pm \psi + g_\pm^{(3),hZ\psi}(\partial_\nu h)Z_\mu \bar{\psi}\gamma^\mu P_\pm(\partial^\nu \psi) \\
&+ g_\pm^{(4),hZ\psi}(\partial_\nu h)(\partial^\nu Z_\mu)\bar{\psi}\gamma^\mu P_\pm \psi + g_\pm^{(5),hZ\psi}(\partial_\nu h)(\partial_\mu Z^\nu)\bar{\psi}\gamma^\mu P_\pm \psi \\
&+ g_\pm^{(6),hZ\psi}(\partial_\nu h)Z_\mu \bar{\psi}\gamma^\nu P_\pm(\partial^\mu \psi) + g_\pm^{(7),hZ\psi}(\partial_\mu \partial_\nu h)Z^\nu \bar{\psi}\gamma^\mu P_\pm \psi \\
&+ g_\pm^{(8),hZ\psi}(\partial_\nu h)Z_\mu(\partial^\mu \bar{\psi})\gamma^\nu P_\pm \psi .
\end{aligned}
\tag{17}
$$

Table 3: Effective couplings of Eqs. 17 and 19, and the effective operators that generate them. For a given $\psi \in \{Q, u, d\}$ the operators above need to be checked against Tabs. 1 and 2. When $V$ appears as a superscript it is to indicate that both the corresponding $B$ and $W$ operators contribute.

| effective vertex | corresponding operators |
|---|---|
| $g_{\pm}^{(4),hZ\psi}$ | $Q_{\psi^2VH^2D}^{(1)}, Q_{\psi^2VH^2D}^{(3)}, Q_{\psi^2H^2D^3}^{(1)}, Q_{\psi^2H^2D^3}^{(3)}$ |
| $g_{\pm}^{(5),hZ\psi}$ | $Q_{\psi^2VH^2D}^{(1)}, Q_{\psi^2VH^2D}^{(3)}, Q_{\psi^2H^2D^3}^{(1)}, Q_{\psi^2H^2D^3}^{(3)}$ |
| $g_{\pm}^{(7),hZ\psi}$ | $Q_{\psi^2H^2D^3}^{(1)}, Q_{\psi^2H^2D^3}^{(3)}$ |
| $g_{L}^{(2),hW^{\pm}\psi}$ | $Q_{\psi^2H^2D^3}^{(4)}$ |
| $g_{L}^{(3),hW^{\pm}\psi}$ | $Q_{\psi^2H^2D^3}^{(4)}$ |
| $g_{L}^{(4),hW^{\pm}\psi}$ | $Q_{\psi^2WH^2D}^{(3)}, Q_{\psi^2WH^2D}^{(5)}, Q_{\psi^2H^2D^3}^{(3)}$ |
| $g_{L}^{(5),hW^{\pm}\psi}$ | $Q_{\psi^2WH^2D}^{(3)}, Q_{\psi^2WH^2D}^{(5)}, Q_{\psi^2H^2D^3}^{(3)}$ |
| $g_{L}^{(6),hW^{\pm}\psi}$ | $Q_{\psi^2H^2D^3}^{(4)}$ |
| $g_{L}^{(7),hW^{\pm}\psi}$ | $Q_{\psi^2H^2D^3}^{(3)}$ |
| $g_{L}^{(8),hW^{\pm}\psi}$ | $Q_{\psi^2H^2D^3}^{(4)}$ |

The $g_{\pm}^{(n),hZ\psi}$ are written in terms of dimension-eight Wilson coefficients in App. A. The subscript $\pm$ is exchanged for $L$ or $R$ when a specific projection is written elsewhere in this article. The full effective Lagrangian of Eq. 17 should be understood as the sum over both projections, $\pm$. As a short summary, Table 3 shows which operators generate each $g$.

## 3.2 $\mathcal{L}_{\text{eff}}$ for charged currents

The charged-current interactions are more subtle as the effective couplings may now be complex. Using the notation introduced above, we find the effective Lagrangian:

$$
\begin{aligned}
\mathcal{L}_{\text{eff}}^{\text{NC},1} = {} & c_{HWW}^{(1)} hW_\mu^+ W_\nu^- + \frac{c_{HWW}^{(2)}}{2} hW_{\mu\nu}W^{\mu\nu} \\
& + c_{HWW}^{(3)}(\partial^\mu h)W_{\mu\nu}^+ W^{-\,\nu} + c_{HWW}^{(4)}(\partial^\mu h)W_{\mu\nu}^- W^{+\,\nu} \\
& + \left( h g_{\pm,pr}^{(1),hW\psi} + g_{\pm,pr}^{W\psi} \right)\bar{\psi}\gamma^\mu P_\pm \psi W_\mu^+ + h.c.
\end{aligned}
\tag{18}
$$

Here, the first two lines correspond to purely bosonic interactions, while the last includes interactions of the $W^\pm$ with both left and right handed fermions. The LH couplings are generated by the renormalizable Lagrangian, while the RH currents are generated first at order $1/\Lambda^2$, as such the right handed currents can only contribute in quadrature (given the assumptions $m_\psi$ is negligible). Expressions for the effective couplings in terms of geometric quantities and expanded out in terms of the Wilson coefficients can be found in App. A. The operators followed by $+h.c.$ have, in general, complex coefficients which are taken into account. As we are assuming a $U(3)^5$ flavor symmetry, $V_{\text{CKM}} \to \delta_{pr}$. A more detailed study of the CKM matrix in the context of the SMEFT and particularly the SMEFT to order $1/\Lambda^4$ is needed to relax this assumption. A strong starting point at dimension-six can be found in [34], while a first look at beyond leading order can be found in [35]. Appendix F discusses the degree to which this assumption is valid by considering Cabibbo suppressed contributions.

Again, we consider the operators of Tables 1 and 2. For the charged currents we find the effective Lagrangian:

$$
\begin{aligned}
\mathcal{L} = {}& g_L^{(2),hW^{\pm}\psi}(\partial_\nu h)W_\mu^{\pm}(\partial_\nu\bar\psi)\gamma^\mu T^{\pm}\psi + g_L^{(3),hW^{\pm}\psi}(\partial_\nu h)W_\mu^{\pm}\bar\psi\gamma^\mu T^{\pm}(\partial^\nu\psi) \\
& + g_L^{(4),hW^{\pm}\psi}(\partial_\nu h)(\partial^\nu W_\mu^{\pm})\bar\psi\gamma^\mu T^{\pm}\psi + g_L^{(5),hW^{\pm}\psi}(\partial^\nu h)(\partial_\mu W_\nu^{\pm})\bar\psi\gamma^\mu T^{\pm}\psi \\
& + g_L^{(6),hW^{\pm}\psi}(\partial_\nu h)W_\mu^{\pm}\bar\psi\gamma^\nu T^{\pm}(\partial^\mu\psi) + g_L^{(7),hW^{\pm}\psi}(\partial_\mu\partial^\nu h)W_\nu^{\pm}\bar\psi\gamma^\mu T^{\pm}\psi \\
& + g_L^{(8),hW^{\pm}\psi}(\partial_\nu h)W_\mu^{\pm}(\partial^\mu\bar\psi)\gamma^\nu T^{\pm}\psi \,,
\end{aligned}
\tag{19}
$$

where $2T^{\pm} = \sigma_1 \pm i\sigma_2$. Considering contributions to $\mathcal{O}(1/\Lambda^4)$ requires $\psi = Q$ as these operators are generated at dimension eight and so the corresponding right-handed operators can only contribute at order $1/\Lambda^6$. The $g_L^{(n),hW^{\pm}\psi}$ are written in terms of dimension-eight Wilson coefficients in App. A. As a short summary, Table 3 shows which operators generate each $g$.

The charged current process was considered in Ref. [4] including $\mathcal{O}(1/\Lambda^4)$ effects, but at the level of a $2 \to 2$ process only. Additionally, the calculation in Ref. [4] was not carried out in a geoSMEFT compliant basis, hence it contains corrections to the $ffW$ vertex originating from $D^3H^2\psi^2$ operators. The results of this work improve upon Ref. [4] by adopting the operator choice in Table 2 and including the effects of the $W^{\pm}$ decay.

# 4 Higgs associated production with a $Z$ and $W^{\pm}$

The matrix elements and their squares were calculated using the Feynrules for the SMEFT found in [36],[4] the FeynArts and FormCalc packages [37,38], as well as cross checked by hand. We consider the case of the 13 TeV LHC, working with NNPDF3.0 NLO parton distribution functions [39,40], $\alpha_s = 0.118$, and a fixed factorization scale $\mu_F = m_Z$.

Details of the phase space and parton distribution function integrations are discussed in App. C. These integrations were separately performed for each combination of couplings in the effective Lagrangians discussed in Sec. 3, as well as the expanded propagator of Sec. 2.3. We can consider the ratio of any given SMEFT cross section to the SM cross section after having factored out all coupling and $N_c$ dependence as a measure of how different the population of phase space is between the SM and the novel kinematics of the SMEFT. This was discussed in the context of the Higgs decay to four fermions at dimension-six in the SMEFT in [32].

In the next two subsections, we present the SM tree-level results as well as these ratios in tables. These combined with the mappings between the SMEFT and our effective Lagrangians allows one to obtain the full expressions for Higgs associated production to order $1/\Lambda^4$ in the SMEFT. As these expressions are cumbersome we do not attempt to write them here, but instead include ancillary files that allow one to fully reproduce them in Mathematica. The last subsection includes some discussion of distributions, further illustrating the differences between the phase space populations in SM and the SMEFT.

## 4.1 Neutral currents

Table 4 lists the primary ingredients for constructing the cross section for $pp \to hZ(\to \bar\ell\ell)$ to order $1/\Lambda^4$ in the SMEFT. In order to fit the full results into the table some simplifications have been made implicitly in the table:

---

[4]In the package we change the field-strength connections so they are general instead of depending on the individual Wilson coefficients.

1. While all integrations were performed to per-mil accuracy, we only display two significant digits. The only exception to this is the first row corresponding to the SM-like contribution.

2. In the SM row we have also factored out a factor of $10^{-4}$ which instead appears with the coupling dependence. We have normalized all the subsequent rows to the SM-like contribution of the first row, including the factor $10^{-4}$, so everything is approximately of the same order. As mentioned above, these quantities have the useful interpretation of expressing the differences in phase-space population of the different kinematics from the SMEFT that contribute to Higgs associated production with a $Z$-boson.

3. We have factored out,

$$\hat{v}^{2n} = \left( \frac{1}{\sqrt{2}\hat{G}_F} \right)^n \sim \left( 246.22 \text{ GeV} \right)^{2n}. \tag{20}$$

From each term corresponding to the leading order in $1/\Lambda^{2n}$ at which the given effective coupling is generated in the SMEFT. For a more straightforward comparison between the expressions we have only factored out $\hat{v}^2$ from the expressions containing $c_{HVV'}^{(3)}$ and $c_{HVV'}^{(4)}$. Although these forms are first generated at order $1/\Lambda^4$ in the Warsaw and geoSMEFT bases, they are generated at $1/\Lambda^2$ in other bases.

Table 4 contains all the necessary results from the phase space and parton distribution function integrations. To the numerical precision included in the table, the $\hat{\alpha}$ and $\hat{m}_W$ schemes produce the same table. Many different coupling combinations lead to the same phase space integrals.[5] Denoting a given amplitude, with coupling dependence removed, by $\mathcal{M}_i$ we have within the SM, for example:

$$\left[ c_{HZZ}^{(1)} \right]^2 \left( g_L^{Z\ell} \right)^2 \left( g_L^{Zq} \right)^2 N_c \int d\sigma |\mathcal{M}_1|^2 = \left[ c_{HZZ}^{(1)} \right]^2 \left( g_R^{Z\ell} \right)^2 \left( g_R^{Zq} \right)^2 N_c \int d\sigma |\mathcal{M}_2|^2, \tag{21}$$

which is to say, for the fully integrated phase space, the difference between the cross section for left- and right-handed couplings of the Z to fermions is fully encoded in the effective couplings $g_{L,R}^{Z\psi}$. In the case above the amplitude squared is identically the same (in the massless limit), however we also find:

$$\left[ c_{HZZ}^{(1)} \right]^2 \left( g_L^{Z\ell} \right)^2 \left( g_L^{Zq} \right)^2 N_c \int d\sigma |\mathcal{M}_1|^2 \simeq \left[ c_{HZZ}^{(1)} \right]^2 \left( g_L^{Z\ell} \right)^2 \left( g_R^{Zq} \right)^2 N_c \int d\sigma |\mathcal{M}_3|^2 \tag{22}$$

$$= \left[ c_{HZZ}^{(1)} \right]^2 \left( g_R^{Z\ell} \right)^2 \left( g_L^{Zq} \right)^2 N_c \int d\sigma |\mathcal{M}_4|^2,$$

where we have used the symbol $\simeq$ to indicate the numerical integrations are the same, however the squared amplitudes are not.[6] Notice that the second line is again an equality as the square amplitudes denoted by $|M_3|^2$ and $|M_4|^2$ are identical. Again, we stress, the numerical "equality" is a result of integrating over the full phase space, and applying cuts can lift this degeneracy. These equivalences occur frequently, as such Table 4 is reduced by taking them into account. Table 5 then lists all these equivalences.

---

[5]This is only true for the fully integrated phase space, they may differ if cuts are included.

[6]Or interference terms, $|\mathcal{M}|^2$ here is simply shorthand for either case.

Given Tables 4 and 5 we can then obtain the full cross section for $pp \rightarrow hZ(\rightarrow \bar{\ell}\ell)$. To clarify this procedure we write out part of the expression for $d\bar{d} \rightarrow HZ(\rightarrow \bar{\ell}\ell)$:

$$
\begin{aligned}
\sigma\left(d\bar{d} \rightarrow hZ(\rightarrow \bar{\ell}\ell)\right) = & \left(\left[c_{HZZ}^{(1)}\right]^2 \left[g_L^{Z\ell}\right]^2 \left[g_L^{Zq}\right]^2 \cdot 10^{-4}\right) \times 2.56 \\
& + \left(c_{HZZ}^{(1)} c_{HZZ}^{(2)} [g_L^{Z\ell}]^2 [g_L^{Zq}]^2 \hat{v}^2\right) \times 0.93 \times (2.56 \cdot 10^{-4}) \\
& + \left(c_{HZZ}^{(1)} c_{HZZ}^{(2)} [g_R^{Z\ell}]^2 [g_R^{Zq}]^2 \hat{v}^2\right) \times 0.93 \times (2.56 \cdot 10^{-4}) \\
& + \left(c_{HZZ}^{(1)} c_{HZZ}^{(2)} [g_R^{Z\ell}]^2 [g_L^{Zq}]^2 \hat{v}^2\right) \times 0.93 \times (2.56 \cdot 10^{-4}) \\
& + \left(c_{HZZ}^{(1)} c_{HZZ}^{(2)} [g_L^{Z\ell}]^2 [g_R^{Zq}]^2 \hat{v}^2\right) \times 0.93 \times (2.56 \cdot 10^{-4}) \\
& + \cdots
\end{aligned}
\tag{23}
$$

The first line of Eq. 23 is just the product of the coupling dependence of the SM Part of Tab. 4 with the entry under $dd$ in the Table. The second line again follows from Tab. 4, namely we take the coupling dependence of the next entry, multiply it by the entry under dd, then remove the SM-like normalization by multiplying by the SM entry from the first row *as well as* the factor of $10^{-4}$ in the coupling part. The next three lines follow from Tab. 5 which tells us the phase space and pdf integrations are the same for each of the different chiral combinations appearing in the middle four lines. The ellipsis then indicates to repeat this procedure for all the subsequent rows in Tab. 4 while taking into account the equivalencies in Tab. 5. As this is a tedious task, and prone to human error, the ancillary files of this publication include a Mathematica notebook with the full expressions for leptons in the final state in terms of the Wilson coefficients. The ancillary files are designed for calculations at fixed order in $1/\Lambda^2$, however with minor alterations an interested reader could modify them to produce only the dimension-six squared contributions at order $1/\Lambda^4$.

Appendix E contains Tab. 10 which is the same as Tab. 4 with the added requirement that the partonic center of mass energy be larger than 500 GeV. Comparing between the two tables we see that the additional kinematic restriction favors the effective vertices which grow with energy. In particular, there is a moderate increase, $\mathcal{O}(2-6)$, in the phase space population for operators such as $c_{HZZ}^{(2,3)}$ and a large increase, $\mathcal{O}(5-20)$, for the contact operators. This comparison can be made using the ancillary files for individual Wilson coefficients. Taking the ratio of the Wilson coefficient of the the operator $Q_i^{(d)}$ for the process with partonic center of mass energy greater than 500 GeV to that with no restriction we find:

$$
\begin{aligned}
c_{H\Box}^{(6)} &\rightarrow 1, \\
c_{HW}^{(6)} &\rightarrow 1.6, \\
c_{Hq}^{(6),1} &\rightarrow 24, \\
c_{q^2H^2D^3}^{(8),3} &\rightarrow 310.
\end{aligned}
\tag{24}
$$

We see operators which simply rescale the SM, such as $Q_{H\Box}^{(6)}$ have no affect, the contribution from operators such as $Q_{HW}^{(6)}$ increases moderately, and the contribution from $Q_{Hq}^{(6),1}$ increases by a factor of about 24. The change in the contribution from dimension-eight contact operators, such as $Q_{q^1H^2D^3}^{(8),3}$, is greatly enhanced due to the additional derivative dependence of the operator. As these processes are better measured in the future, this may allow for discrimination between the different contributions from the SMEFT at dimensions six and eight. In

order to take advantage of this, a natural extension of this work is to support experimental searches and understand how to move from SM searches to searches which focus on the novel kinematics of the SMEFT.

## 4.2 Charged currents

The charged currents proceed similarly to the neutral currents. There are some notable differences, however. They can be summed up as:

1. The effective couplings may be complex (See Eq. 18 and App. A). Therefore our expressions depend on the real and imaginary parts of effective coupling combinations. Under our $U(3)^5$ assumption the CKM matrix is taken to be the identity matrix. This is corrected by SMEFT contributions, however, and therefore the SM part of the coupling $W^{\pm}\bar{\psi}\psi$ is real, but there is a $1/\Lambda^2$ correction that generates an imaginary part.

2. The $W$s only couple to left handed fermions in the renormalizable Lagrangian, so the number of relations between amplitudes is reduced, but the number we must write out is increased due to the first point above. As the right-handed couplings are generated at different order from the left-handed couplings we explicitly write them in the table (i.e. we do not write out all the relations between combinations as in Tab. 5.

3. We must consider final states $W^+$ and $W^-$ which correspond to different combinations of parton distribution functions.

As with the neutral currents example, Table 6 presents our main results for the $\hat{\alpha}$ scheme. Table 9 in App. E contains the results for the $\hat{m}_W$ scheme. As was the case with the $Z$ we see interesting differences of the phase space population in the SMEFT relative to just the SM. This is also the first time $m_W$ affects our results and therefore $\delta m_W$ is present in the table. Furthermore, that the $W$ couplings are complex means our results depend on the real and imaginary parts of effective couplings and/or combinations of effective couplings. In the case of the contribution of imaginary parts, we generally find the results are strongly suppressed compared with the real part. Imaginary parts of the effective couplings only show up at $\mathcal{O}(1/\Lambda^4)$ (see Appendix A) and can therefore only contribute to the order we have calculated by interfering with imaginary parts of the SM amplitude. The result is that imaginary coefficients always multiply $\Gamma_W$ or a Levi-Civita contraction of momenta; the former are suppressed by $\Gamma_W \ll m_W, \hat{s}$, while the latter is zero upon integration over the full phase space.

With the information above we can construct the full production cross section from Table 6, just as in Eq. 23:

$$
\sigma\left(u\bar{d} \to W^+ h\right) = \left([c_{HWW}^{(1)}]^2 |g_L^{W\ell}|^2 |g_L^{Wq}|^2 \cdot 10^{-4}\right) \times 5.74
$$

$$
+ \left([c_{HWW}^{(2)}]^2 |g_L^{W\ell}|^2 |g_L^{Wq}|^2\right) \times 0.30 \times (5.74 \cdot 10^{-4})
$$

$$
+ \left(c_{HWW}^{(1)} c_{HWW}^{(2)} |g_L^{W\ell}|^2 |g_L^{Wq}|^2\right) \times 0.78 \times (5.74 \cdot 10^{-4}) \tag{25}
$$

$$
+ \cdots
$$

Again the ellipses indicates repeating the above for the remaining row of Tab. 6. Adding the $c\bar{s}$ column gives the full result of $W^+$ production in association with a Higgs boson. Then in order to obtain production of a $W^-$ we repeat this procedure for the last two columns of the table.

Table 4: The first column is the coupling dependence which has been removed from the cross section normalized to the SM-like cross section (again stripped of coupling dependence) for each parton combination that follows in the other columns. In the first row, the SM-like contribution to the cross section is not normalized as in the other cases. We note a factor of $10^{-4}$ has been factored out of the SM-like row. For the other rows a factor of $\hat{v}^{2n}$ has been added based on the leading order at which the contribution can be generated. In the specific case of $c_{HZZ}^{(3)}$ and $c_{HAZ}^{(3)}$ this is taken to be $\hat{v}^2$, although in the Warsaw and geoSMEFT bases these operators first occur at dimension-eight. The coupling dependence in the first column should be multiplied by the constants to the right as well as the SM-like constants of the first row in order to obtain the full contribution to the cross section for the given effective coupling dependence. To the numerical precision displayed in the table, the $\hat{\alpha}$ and $\hat{m}_W$ schemes do not differ.

| | partons | | | | |
|---|---|---|---|---|---|
| ($\mathcal{L}_{\text{eff}}$ dependence)$/N_c$ | dd | uu | ss | cc | bb |
| $[c_{HZZ}^{(1)}]^2 [g_L^{Z\ell}]^2 [g_L^{Zq}]^2 \cdot 10^{-4}$ | 2.56 | 3.86 | 0.559 | 0.311 | 0.122 |
| $c_{HZZ}^{(1)} c_{HZZ}^{(2)} [g_L^{Z\ell}]^2 [g_L^{Zq}]^2 \hat{v}^2$ | 0.93 | 0.94 | 0.90 | 0.89 | 0.88 |
| $c_{HZZ}^{(1)} c_{HZZ}^{(3)} [g_L^{Z\ell}]^2 [g_L^{Zq}]^2 \hat{v}^2$ | 2.1 | 2.2 | 1.7 | 1.5 | 1.5 |
| $c_{HAZ}^{(2)} c_{HZZ}^{(1)} \bar{e} [g_L^{Z\ell}]^2 g_L^{Zq} Q_q \hat{v}^2$ | -0.84 | -0.85 | -0.81 | -0.80 | -0.79 |
| $c_{HAZ}^{(3)} c_{HZZ}^{(1)} \bar{e} [g_L^{Z\ell}]^2 g_L^{Zq} Q_q \hat{v}^2$ | -2.6 | -2.7 | -2.1 | -2.0 | -1.9 |
| $[c_{HZZ}^{(2)}]^2 [g_L^{Z\ell}]^2 [g_L^{Zq}]^2 \hat{v}^4$ | 0.38 | 0.39 | 0.30 | 0.28 | 0.28 |
| $[c_{HAZ}^{(2)}]^2 \bar{e}^2 [g_L^{Z\ell}]^2 Q_q^2 \hat{v}^4$ | 0.66 | 0.69 | 0.52 | 0.48 | 0.47 |
| $c_{HAZ}^{(2)} c_{HZZ}^{(2)} \bar{e} [g_L^{Z\ell}]^2 g_L^{Zq} Q_q \hat{v}^4$ | -0.70 | -0.73 | -0.56 | -0.52 | -0.51 |
| $c_{HZZ}^{(1)} [g_L^{Z\ell}]^2 g_L^{Zq} g_L^{(1),hZq} \hat{v}^2$ | 3.4 | 3.5 | 2.9 | 2.8 | 2.7 |
| $[g_L^{Z\ell}]^2 [g_L^{(1),hZq}]^2 \hat{v}^4$ | 8.9 | 11 | 4.6 | 3.6 | 3.3 |
| $c_{HZZ}^{(1)} [g_L^{Z\ell}]^2 g_L^{Zq} g_L^{(4),hZq} \hat{v}^4$ | 8.5 | 10 | 4.2 | 3.2 | 3.0 |
| $c_{HZZ}^{(1)} [g_L^{Z\ell}]^2 g_L^{Zq} g_L^{(5),hZq} \hat{v}^4$ | 7.0 | 8.6 | 3.1 | 2.2 | 2.0 |
| $c_{HZZ}^{(1)} [g_L^{Z\ell}]^2 g_L^{Zq} g_L^{(7),hZq} \hat{v}^4$ | -7.0 | -8.5 | -3.1 | -2.2 | -2.0 |
| $c_{HAZ}^{(2)} \bar{e} [g_L^{Z\ell}]^2 g_L^{(1),hZq} Q_q \hat{v}^4$ | -1.8 | -1.9 | -1.4 | -1.3 | -1.3 |
| $c_{HZZ}^{(2)} [g_L^{Z\ell}]^2 g_L^{Zq} g_L^{(1),hZq} \hat{v}^4$ | 1.9 | 2.0 | 1.5 | 1.4 | 1.4 |
| $[c_{HZZ}^{(1)}]^2 [g_L^{Z\ell}]^2 [g_L^{Zq}]^2 \delta\Gamma$ | -0.41 | -0.41 | -0.41 | -0.41 | -0.41 |
| $[c_{HZZ}^{(1)}]^2 [g_L^{Z\ell}]^2 [g_L^{Zq}]^2 \delta\Gamma^2$ | 0.17 | 0.17 | 0.17 | 0.17 | 0.17 |
| $c_{HZZ}^{(1)} c_{HZZ}^{(2)} [g_L^{Z\ell}]^2 [g_L^{Zq}]^2 \delta\Gamma \hat{v}^2$ | -0.39 | -0.39 | -0.37 | -0.37 | -0.37 |
| $c_{HAZ} c_{HZZ}^{(1)} \bar{e} [g_L^{Z\ell}]^2 g_L^{Zq} Q_q \delta\Gamma \hat{v}^2$ | 0.35 | 0.35 | 0.34 | 0.33 | 0.33 |
| $c_{HZZ}^{(1)} [g_L^{Z\ell}]^2 g_L^{Zq} g_L^{(1),hZq} \delta\Gamma \hat{v}^2$ | -1.4 | -1.5 | -1.2 | -1.1 | -1.1 |

Table 5: Redundant phase space integrations categorized by the coupling constants corresponding to each integral as discussed around Eq. 21. The symbols "=" and "≃" should be understood as discussed in the main text. $i$ and $j$ as superscripts should be understood for $i, j \in \{1, 2, \cdots\}$ consistent with the Effective Lagrangians of Sec. 3. These relations have only been tested to order $1/\Lambda^4$, and so should not be extrapolated beyond this order.

$$
\begin{aligned}
\left[c_{HZZ}^{(i)}\right]^2 \left[g_L^{Z\ell}\right]^2 \left[g_L^{Zq}\right]^2 &= \left[c_{HZZ}^{(i)}\right]^2 \left[g_R^{Z\ell}\right]^2 \left[g_R^{Zq}\right]^2 \\
&\simeq \left[c_{HZZ}^{(i)}\right]^2 \left[g_L^{Z\ell}\right]^2 \left[g_R^{Zq}\right]^2 \\
&= \left[c_{HZZ}^{(i)}\right]^2 \left[g_R^{Z\ell}\right]^2 \left[g_L^{Zq}\right]^2 \\[4pt]
\hline
c_{HZZ}^{(1)} c_{HZZ}^{(j)} \left[g_L^{Z\ell}\right]^2 \left[g_L^{Zq}\right]^2 &= c_{HZZ}^{(1)} c_{HZZ}^{(j)} \left[g_R^{Z\ell}\right]^2 \left[g_R^{Zq}\right]^2 \\
&\simeq c_{HZZ}^{(1)} c_{HZZ}^{(j)} \left[g_L^{Z\ell}\right]^2 \left[g_R^{Zq}\right]^2 \\
&= c_{HZZ}^{(1)} c_{HZZ}^{(j)} \left[g_R^{Z\ell}\right]^2 \left[g_L^{Zq}\right]^2 \\[4pt]
\hline
\left[c_{HAZ}^{(2)}\right]^2 \bar{e}^2 \left[g_R^{Z\ell}\right]^2 Q_q^2 &= \left[c_{HAZ}^{(2)}\right]^2 \bar{e}^2 \left[g_L^{Z\ell}\right]^2 Q_q^2 \\[4pt]
\hline
c_{HAZ}^{(i)} c_{HZZ}^{(j)} \bar{e} \left[g_L^{Z\ell}\right]^2 g_L^{Zq} Q_q &= c_{HAZ}^{(i)} c_{HZZ}^{(j)} \bar{e} \left[g_R^{Z\ell}\right]^2 g_R^{Zq} Q_q \\
&\simeq c_{HAZ}^{(i)} c_{HZZ}^{(j)} \bar{e} \left[g_R^{Z\ell}\right]^2 g_L^{Zq} Q_q \\
&= c_{HAZ}^{(i)} c_{HZZ}^{(j)} \bar{e} \left[g_L^{Z\ell}\right]^2 g_R^{Zq} Q_q \\[4pt]
\hline
\left[g_L^{(1),hZq}\right]^2 \left[g_L^{Z\ell}\right]^2 &= \left[g_R^{(1),hZq}\right]^2 \left[g_R^{Z\ell}\right]^2 \\
&\simeq \left[g_L^{(1),hZq}\right]^2 \left[g_R^{Z\ell}\right]^2 \\
&= \left[g_R^{(1),hZq}\right]^2 \left[g_L^{Z\ell}\right]^2 \\[4pt]
\hline
c_{HZZ}^{(i)} g_L^{(i),hZq} \left[g_L^{Z\ell}\right]^2 g_L^{Zq} &= c_{HZZ}^{(i)} g_R^{(i),hZq} \left[g_R^{Z\ell}\right]^2 g_R^{Zq} \\
&\simeq c_{HZZ}^{(i)} g_L^{(i),hZq} \left[g_R^{Z\ell}\right]^2 g_L^{Zq} \\
&= c_{HZZ}^{(i)} g_R^{(i),hZq} \left[g_L^{Z\ell}\right]^2 g_R^{Zq} \\[4pt]
\hline
c_{HAZ}^{(1)} g_L^{(1),hZq} \left[g_L^{Z\ell}\right]^2 g_L^{Zq} &= c_{HAZ}^{(1)} g_R^{(1),hZq} \left[g_R^{Z\ell}\right]^2 g_R^{Zq} \\
&\simeq c_{HAZ}^{(1)} g_L^{(1),hZq} \left[g_R^{Z\ell}\right]^2 g_L^{Zq} \\
&= c_{HAZ}^{(1)} g_R^{(1),hZq} \left[g_L^{Z\ell}\right]^2 g_R^{Zq}
\end{aligned}
$$

## 4.3 Distributions

The integration procedure used to calculate the inclusive cross section in the previous sections can easily be adjusted to calculate kinematic distributions by mimicking MadGraph's [41] reweight [42] method in our Monte Carlo.

Specifically, for each phase space point $z_i$ generated via Monte Carlo, rather than just output the weight for that point for one matrix element, we output the weight for every matrix element type (all entries in Table 4 or Table 6) along with the initial and final four vectors. Combining the weights for the different matrix elements with their corresponding couplings – the leftmost column of Table 4 or 6 expanded out to $\mathcal{O}(\hat{v}^2/\Lambda^2)$ – allows us to obtain the

Table 6: Charged current phase space and pdf integral table for the $\hat{\alpha}$ scheme. The asterisk for certain table entries are marked as they have an extra factor of $10^{-x}$, for some $x$, in the first column, and so should not be read to be of a similar size to the other entries in the table. Again we include factors of $\hat{v}$ in the first column consistent with the order at which a given effective vertex is generated in the SMEFT. Note, however, that we do not do this in the case of the right handed coupling. In general $g_L^{W\psi}$ is complex, however under our assumptions the imaginary part enters at $\mathcal{O}(1/\Lambda^2)$. As such, when $g_L^{W\psi}$ occurs in the table and is not written as $|g_L^{W\psi}|$ or the real/imaginary part is not explicitly taken, we are implicitly using the SM part.

| | partons | | | |
|---|:---:|:---:|:---:|:---:|
| $(\mathcal{L}_{\text{eff}}$ dependence$)/N_c$ | $u\bar{d}$ | $c\bar{s}$ | $\bar{u}d$ | $\bar{c}s$ |
| $[c_{HWW}^{(1)}]^2 \lvert g_L^{W\ell}\rvert^2 \lvert g_L^{Wq}\rvert^2 \cdot 10^{-4}$ | 5.74 | 0.577 | 3.36 | 0.637 |
| $[c_{HWW}^{(2)}]^2 \lvert g_L^{W\ell}\rvert^2 \lvert g_L^{Wq}\rvert^2 \hat{v}^4$ | 0.30 | 0.21 | 0.28 | 0.22 |
| $c_{HWW}^{(1)} c_{HWW}^{(2)} \lvert g_L^{W\ell}\rvert^2 \lvert g_L^{Wq}\rvert^2 \hat{v}^2$ | 0.78 | 0.75 | 0.78 | 0.75 |
| $c_{HWW}^{(1)} c_{HWW}^{(3)} \lvert g_L^{W\ell}\rvert^2 \lvert g_L^{Wq}\rvert^2 \hat{v}^2$ | 0.57 | 0.53 | -2.7 | -2.2 |
| $c_{HWW}^{(1)} c_{HWW}^{(4)} \lvert g_L^{W\ell}\rvert^2 \lvert g_L^{Wq}\rvert^2 \hat{v}^2$ | -2.9 | -2.1 | 0.56 | 0.54 |
| $[c_{HWW}^{(1)}]^2 \lvert g_L^{W\ell}\rvert^2 \lvert g_R^{Wq}\rvert^2$ | 5.74 | 0.577 | 3.36 | 0.637 |
| $\lvert g_L^{W\ell}\rvert^2 \lvert g_L^{(1),hWq}\rvert^2 \hat{v}^4$ | 12 | 3.5 | 8.2 | 4.1 |
| $c_{HWW}^{(1)} \lvert g_L^{W\ell}\rvert^2 \mathrm{Re}[(g_L^{(1),hWq})^* g_L^{Wq}]\hat{v}^2$ | 1.7 | 1.3 | 1.6 | 1.4 |
| $c_{HWW}^{(2)} \lvert g_L^{W\ell}\rvert^2 \mathrm{Re}[(g_L^{(1),hWq})^* g_L^{Wq}]\hat{v}^2$ | 1.68 | 1.16 | 1.54 | 1.21 |
| $c_{HWW}^{(1)} \lvert g_L^{W\ell}\rvert^2 \mathrm{Re}[(g_L^{(4),hWq})^* g_L^{Wq}]\hat{v}^4$ | 5.1 | 1.6 | 3.9 | 1.9 |
| $c_{HWW}^{(1)} \lvert g_L^{W\ell}\rvert^2 \mathrm{Re}[(g_L^{(5),hWq})^* g_L^{Wq}]\hat{v}^4$ | 4.5 | 1.1 | 3.3 | 1.4 |
| $c_{HWW}^{(1)} \lvert g_L^{W\ell}\rvert^2 \mathrm{Im}[(g_L^{(1),hWq})^* g_L^{Wq}]\hat{v}^2 \cdot 10^{-3}$ | 2.7* | 2.6* | -2.7* | -2.7* |
| $c_{HWW}^{(1)} \lvert g_L^{W\ell}\rvert^2 \mathrm{Im}[(g_L^{(4),hWq})^* g_L^{Wq}]\hat{v}^4 \cdot 10^{-3}$ | 2.0* | 1.4* | -1.8* | -1.5* |
| $c_{HWW}^{(1)} \lvert g_L^{W\ell}\rvert^2 \mathrm{Im}[(g_L^{(5),hWq})^* g_L^{Wq}]\hat{v}^4 \cdot 10^{-3}$ | 1.2* | 0.71* | -1.1* | -0.75* |
| $c_{HWW}^{(1)} \lvert g_L^{W\ell}\rvert^2 g_L^{(2),hWq} g_L^{Wq} \hat{v}^4$ | -2.8 | -0.95 | 2.1 | 1.1 |
| $c_{HWW}^{(1)} \lvert g_L^{W\ell}\rvert^2 g_L^{(3),hWq} g_L^{Wq} \hat{v}^4$ | -2.8 | -0.95 | -2.1 | -1.1 |
| $c_{HWW}^{(1)} \lvert g_L^{W\ell}\rvert^2 g_L^{(6),hWq} g_L^{Wq} \hat{v}^4$ | 2.3 | 0.57 | 1.6 | 0.72 |
| $c_{HWW}^{(1)} \lvert g_L^{W\ell}\rvert^2 g_L^{(7),hWq} g_L^{Wq} \hat{v}^4$ | -4.5 | -1.1 | -3.3 | -1.4 |
| $c_{HWW}^{(1)} \lvert g_L^{W\ell}\rvert^2 g_L^{(8),hWq} g_L^{Wq} \hat{v}^4$ | 2.3 | 0.57 | 1.6 | 0.71 |
| $[c_{HWW}^{(1)}]^2 \lvert g_L^{W\ell}\rvert^2 \lvert g_L^{Wq}\rvert^2 \delta\Gamma$ | -0.50 | -0.50 | -0.50 | -0.50 |
| $[c_{HWW}^{(1)}]^2 \lvert g_L^{W\ell}\rvert^2 \lvert g_L^{Wq}\rvert^2 \delta\Gamma^2$ | 0.25 | 0.25 | 0.25 | 0.25 |
| $c_{HWW}^{(1)} c_{HWW}^{(2)} \lvert g_L^{W\ell}\rvert^2 \lvert g_L^{Wq}\rvert^2 \delta\Gamma \hat{v}^2$ | -0.39 | -0.38 | -0.39 | -0.38 |
| $[c_{HWW}^{(1)}]^2 \lvert g_L^{W\ell}\rvert^2 \lvert g_L^{Wq}\rvert^2 \delta M$ | -0.013 | -0.016 | -0.014 | -0.016 |
| $[c_{HWW}^{(1)}]^2 \lvert g_L^{W\ell}\rvert^2 \lvert g_L^{Wq}\rvert^2 \delta M \delta\Gamma \cdot 10^{-3}$ | 6.7* | 8.1* | 6.9* | 7.9* |
| $[c_{HWW}^{(1)}]^2 \lvert g_L^{W\ell}\rvert^2 \lvert g_L^{Wq}\rvert^2 \delta M^2 \cdot 10^{-4}$ | 1.2* | 1.5* | 1.1* | 1.4* |
| $c_{HWW}^{(1)} c_{HWW}^{(2)} \lvert g_L^{W\ell}\rvert^2 \lvert g_L^{Wq}\rvert^2 \delta M \hat{v}^2 \cdot 10^{-3}$ | 3.3* | 0.85* | 2.8* | 0.11* |
| $c_{HWW}^{(1)} \lvert g_L^{W\ell}\rvert^2 \mathrm{Re}[(g_L^{(1),hWq})^*] g_L^{Wq} \delta M \hat{v}^2$ | -0.017 | -0.015 | -0.016 | -0.015 |
| $c_{HWW}^{(1)} \lvert g_L^{W\ell}\rvert^2 \mathrm{Re}[(g_L^{(1),hWq})^*] g_L^{Wq} \delta\Gamma \hat{v}^2$ | -0.87 | -0.67 | -0.81 | -0.69 |

weight for each event as a function of the Wilson coefficients, $\frac{\hat{v}}{\Lambda}$, and inputs like $\hat{e}$, $\hat{v}$, etc, which we can use to form weighted histograms of whatever parton-level kinematic variables we like (and impose parton-level cuts). More precisely, each event gets a weight

$$w_{\text{SMEFT},q\bar{q}}(z_i) = d\Phi \left( f_q(x_1) f_{\bar{q}}(x_2) + x_1 \leftrightarrow x_2 \right) |c_{1,\text{SM}}^{q\bar{q}}|^2 |\mathcal{M}_1(z_i)|^2 \sum_{j=1}^{n} \frac{|c_{j,\text{SMEFT}}^{q\bar{q}}|^2 |\mathcal{M}_j(z_i)|^2}{|c_{1,\text{SM}}^{q\bar{q}}|^2 |\mathcal{M}_1(z_i)|^2},$$

where $j = 1$ corresponds to the SM matrix element and the number $n$ of matrix elements we include depends on the EW input scheme and whether we are looking at charged current or neutral current events. The factor $d\Phi$ includes normalization/phase space/flux factors, and $f_q, f_{\bar{q}}$ are the parton distribution functions; these factors cancel in the ratio in the sum. The coupling factors carry a dependence on the parton distribution function $(q, \bar{q})$ as the Wilson coefficients (and $\gamma/Z$ couplings) are different for up and down type fermions, while the $\mathcal{M}_j$ are purely kinematic and therefore are insensitive to the parton flavor. Note that, under our $U(3)^5$ assumption, the only flavor dependence carried by the Wilson coefficients is up-type quarks vs. down-type.

For each of the final states, $Z(\ell^+\ell^-)H$, $W^\pm(\ell^\pm\nu)H$, we show two distributions below; $p_{T,\ell\ell}$, $p_{T,\ell^-}$ for the $Z(\ell^+\ell^-)H$ channel, and $p_{T,W^+}$, $\not{E}_T$ for $W^\pm(\ell^\pm\nu)H$. The vector boson transverse momenta are the key ingredient in how associated production events are binned in the STXS Higgs analyses [43], while the individual lepton/neutrino properties are an example of something we need the full $2 \to 3$ machinery to study. Note that, to make distributions for individual lepton/neutrino properties we can no longer assume the equivalence among different helicity contributions so we need to generate events/weights for all combinations.

Focusing on $pp \to W^+(\ell^+\nu)H$ first, there are many different operators involved, so we will explore the distributions by turning on operators/coefficients one at a time.[7] From our discussion in Sec. 3, the operators fall into three classes: i.) geoSMEFT contributions to the $hVV$ vertex, ii.) geoSMEFT contributions to the $ffV$ vertex, and iii.) contact terms. In Fig. 2 below we show the $p_{T,W}$ and $p_{T,\ell}$ distributions when representative coefficients from each of the three classes is turned on and all other operators are zero. Specifically, the four operators we turn on are $c_{HW}^{(6)}$, representing geoSMEFT $hVV$ effects, $c_{HQ}^{(6),3}$ representing geoSMEFT $ffV$ terms, and $c_{Q^2H^2D^3}^{(8),3}$ $c_{Q^2WH^2D}^{(8),3}$ for contact terms. The nonzero coefficient in each case is set to $+1$ with $\Lambda = 3\,\text{TeV}$.[8] In order to zoom in on the phase space regions where SMEFT effects have a larger impact, we impose a parton-level cut of $p_{T,V} > 150\,\text{GeV}$, where $V$ is the vector boson reconstructed from its decay products.

All of the operators we have selected generate new vertices and potentially carry kinematics (momentum dependence) that is distinct from the SM. This should be contrasted with operators that are only involved in parameter definitions, which carry no momentum dependence so all their effects are $\sim \hat{v}^2/\Lambda^2$. To emphasize different SMEFT kinematics rather than changes in the overall distribution normalization, we have plotted normalized distributions.

The impact of the different operators in Fig. 2 is set (up to accidental cancellations) by the operator mass dimension and whether the gauge boson $W/Z$ in the operator arises from a $D_\mu H$ type term, and is therefore predominantly longitudinally polarized, or a $W_{\mu\nu}^I$ type term, in which case the gauge boson is predominantly transverse. The operator mass dimension is

---

[7]For the remainder of this section, we will abuse notation slightly and refer to operators and their Wilson coefficients synonymously, e.g saying 'the operator $c_{HW}^{(6)}$' for brevity rather than 'the operator with Wilson coefficient $c_{HW}^{(6)}$'.

[8]While we have generated the distributions assuming $\Lambda = 3\,\text{TeV}$, the relevant quantity is $c_i^{(6)}/\Lambda^2$, $c_i^{(8)}/\Lambda^4$ with $c_i^{(6,8)}$ the Wilson coefficient of a dimension six or eight operator, so the above plots translate to other $\Lambda$ values with coefficients $c_i^{(6)}(\Lambda/3\,\text{TeV})^2$, $c_i^{(8)}(\Lambda/3\,\text{TeV})^4$. Keeping $c_i^{(6,8)}$ fixed and varying $\Lambda$ has the expected result – SMEFT deviations growing for $\Lambda < 3\,\text{TeV}$ and shrinking for $\Lambda > 3\,\text{TeV}$.

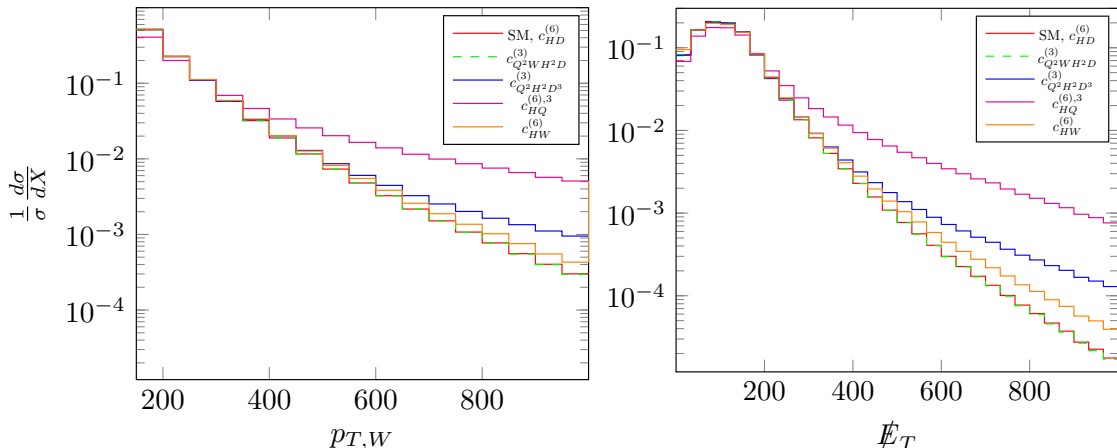

Figure 2: Normalized distributions for $p_{T,W}$ (left panel) and $\not{E}_T$ (right panel) in SMEFT versus in the SM for different Wilson coefficients set to $+1$ while all others taken to be zero, $\Lambda = 3$ TeV. $X$ on the vertical axis refers to the kinematic quantity on each horizontal axis. A dashed line is used for $c^{(3)}_{Q^2 W H^2 D} = 1$ as this case is nearly indistinguishable from the SM until the highest energy bins shown.

important as dimension six operators can interfere with the SM at $\mathcal{O}(1/\Lambda^2)$ and have a 'self-square' contribution at $\mathcal{O}(1/\Lambda^4)$, while dimension eight operators contribute at $\mathcal{O}(1/\Lambda^4)$ only through interference. Interference with the SM not only restricts which operators can enter – as it requires matching the SM helicity/color/polarization structure – but it also introduces factors of small SM couplings. The origin of the gauge boson ($D_\mu H$ vs. $W^I_{\mu\nu}$) is relevant as the SM amplitude for $pp \to VH, V = W^\pm/Z$ is largest for longitudinally polarized $V$, $\mathcal{O}(1)$ vs. $\mathcal{O}(\hat{v}/\sqrt{s})$ for transversely polarized.

Looking first at the dimension six operators in Fig. 2, $c^{(6),3}_{HQ}$ has a much larger effect than $c^{(6)}_{HW}$. While $c^{(6),3}_{HQ}$ is part of the geoSMEFT metric $L^Q_{JA}$ and therefore shifts the $ffV$ couplings from their SM values, it also generates a $\bar{q}qWh$ contact term. The latter is the cause of the large deviation from the SM, as it leads to an energetically enhanced $pp \to W^\pm H$ amplitude for longitudinally polarized $W^\pm$ (the operator contains no field strengths, so the $W^\pm$ must come from $D_\mu H$). This leads to net $pp \to W^\pm H$ amplitude-squared terms of $\mathcal{O}(\hat{s}/\Lambda^2)$ from interference with the SM, and a self-squared term $\mathcal{O}(\hat{s}^2/\Lambda^4)$. The other dimension six operator, $c^{(6)}_{HW}$, does introduce novel momentum dependence into the $hVV$ vertex, though it primarily contributes to $pp \to W^\pm H$ amplitudes with transversely polarized $W^\pm$. The mismatch with the dominant SM polarization suppresses the interference term. The $c^{(6)}_{HW}$ self-square term is unaffected by the polarization mismatch, but it leads to slower growth in $pp \to W^\pm H$, $\sim \mathcal{O}(\hat{s}\hat{v}^2/\Lambda^4)$.

Turning to the dimension-eight operators, $c^{(8),3}_{Q^2 H^2 D^3}$ generates an amplitude that grows as $\hat{s}^2/\Lambda^4$ for $pp \to W^\pm H$ with the $W^\pm$ being longitudinally polarized. This interferes with the dominant SM polarization, leading to a net $\mathcal{O}(\hat{s}^2/\Lambda^4)$ amplitude squared. While this is the same energy dependence as the $c^{(6),3}_{HQ}$ self-square term, the $c^{(8),3}_{Q^2 H^2 D^3}$ piece, being an interference term, is accompanied by additional SM coupling factors. For the particular case here, the extra factors amount to an $\mathcal{O}(40)$ numerical suppression for $c^{(8),3}_{Q^2 H^2 D^3}$ relative to $|c^{(6),3}_{HQ}|^2$.

The final operator, $c^{(8),3}_{Q^2 H^2 XD}$, has very little effect. This is due to the fact that $c^{(8),3}_{Q^2 H^2 XD}$ must interfere with the SM to contribute at $\mathcal{O}(1/\Lambda^4)$ but their polarizations are mismatched. By this we mean that the energetically enhanced amplitude from $c^{(8),3}_{Q^2 H^2 XD}$ involves transversely

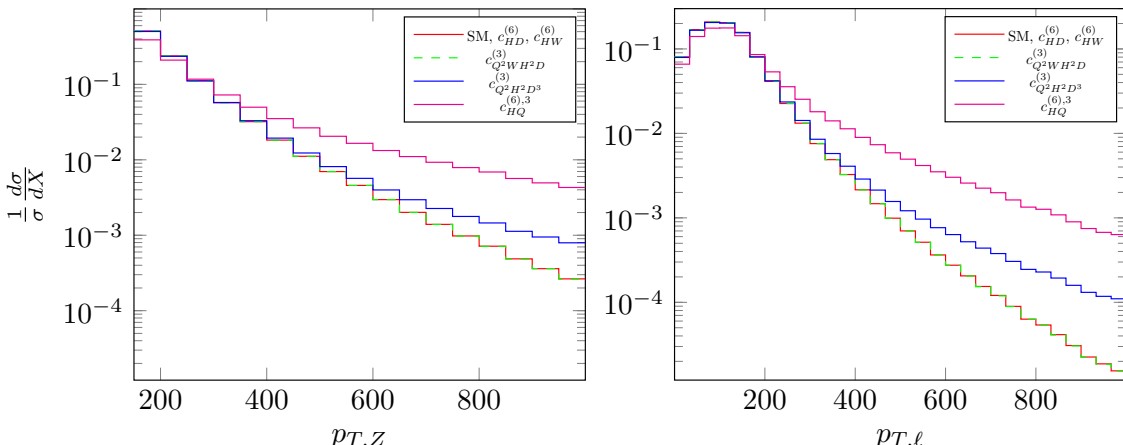

Figure 3: Normalized distributions for $p_{T,Z}$ and $p_{T,\ell}$ for different Wilson coefficients set to +1 while all others taken to be zero, $\Lambda = 3$ TeV. $X$ on the vertical axis refers to the kinematic quantity on each horizontal axis. A dashed line is used for $c_{Q^2WH^2D}^{(3)} = 1$ as this case is nearly indistinguishable from the SM until the highest energy bins shown. $c_{HW}^{(6)}$ is labelled by the red line, this is because it is barely distinguishable for the entirety of the distribution until the last bins where there is a deviation which is nearly not visible on the plot.

polarized $W^{\pm}$, while this polarization is suppressed in the SM by $\mathcal{O}(\hat{v}/\sqrt{s})$. The net effect in the $pp \to W^{\pm}H$ amplitude squared is $\mathcal{O}(\hat{s}\hat{v}^2/\Lambda^4)$, the same order as $|c_{HW}^{(6)}|^2$, though numerically smaller due to additional powers of SM couplings.

Stacking the effects together, the deviation between the SMEFT and the SM is completely driven by $c_{HQ}^{(6),3}$. This, however, assumes that we take all coefficients to be the same size. As $c_{HQ}^{(6),3}$ causes the $ffZ$ couplings to shift from their SM values, it – along with analogous coefficients for other fermions such as $c_{HL}^{(6),3}$, $c_{Hu}^{(6)}$, $c_{Hd}^{(6)}$, etc. – is highly constrained by precision electroweak physics from LEP [44]. For example, the global fit bounds (individual operator, not marginalized bound) from Ref. [3, 45] restrict $|c_{HQ}^{(6),3}| \sim 0.1$ for our choice of $\Lambda$. These bounds were determined ignoring dimension-eight effects, however, we expect that including them (e.g. including $c_{HQ}^{(8),3}$ into the fit) will have little impact as the relevant observables come from resonant $Z$ production where all higher dimensional effects are suppressed by powers of $\hat{v}^2/\Lambda^2$.[9] Implementing this bound on $c_{HQ}^{(6),3}$, the difference between SMEFT and the SM at high energy would be driven by the dimension eight contact terms.

Moving to $pp \to Z^{\pm}(\ell^+\ell^-)H$, the distributions for $p_{T,Z}$ and $p_{T,\ell^-}$ are shown below in Fig. 3. We follow the same procedure as in $pp \to W^+(\ell^+\nu)H$, setting one representative coefficient to +1, $\Lambda = 3$ TeV while all others are zero. As with the previous figures, we impose a parton level cut of $p_{T,\ell\ell} > 150$ GeV to highlight the region where the SMEFT and SM differ. The $Z(\ell^+\ell^-)H$ final state is more involved than $W^{\pm}(\ell^{\pm}\nu)H$ as multiple helicities and parton distribution functions contribute (this was seen for Drell-Yan production in Ref. [10]), however we avoid some of this by turning on only one coefficient at a time. For simplicity, we keep the same representative operators as in $W^+(\ell^+\nu)H$.

The same trends that were present in Fig. 2 show up for $Z^{\pm}(\ell^+\ell^-)H$. With all coefficients

---

[9]On resonance, the dominant corrections come from three-point vertices, which have no non-trivial momentum dependence thus $\hat{v}$ is the only dimensionful scale around. Direct four-point contributions, e.g. to $e^+e^- \to f\bar{f}$ are further suppressed by factors of $\Gamma_Z/m_Z$.

taken equal, $c_{HQ}^{(6),3}$ has the largest effect, due to the fact that its contact term enters at $\mathcal{O}(1/\Lambda^2)$ and that its $\mathcal{O}(1/\Lambda^4)$ self-square contribution is not suppressed by additional SM couplings. The biggest difference between Fig. 3 and Fig. 2 lies in the $c_{HW}^{(6)}$ contribution. Though the energy parametrics for this piece are the same in both processes, the numerical prefactor for $|c_{HW}^{(6)}|$ is significantly smaller for $pp \to Z^\pm(\ell^+\ell^-)H$.

Finally, we have generated the above plots assuming all coefficients are positive. Choosing $c_i < 0$, the interference pieces switch sign.[10] Furthermore, playing with the relative signs between coefficients, one could arrange conspiracies leading to larger SMEFT deviations than we have shown. The deviations we show should therefore be viewed as rough illustrations of which types of operators are most important when it comes to generating kinematics that are different from the SM.

## 5 Conclusions

Using the geoSMEFT approach supplemented by the relevant operators at dimension-eight we have derived the inclusive cross section for Higgs associated production with a $W^\pm$ or $Z$ boson at $\mathcal{O}(1/\Lambda^4)$ in the SMEFT expansion (and leading order in SM couplings). In order to achieve this we identified all operators at dimension eight inconsistent with the geoSMEFT formulation, this has the further consequence of simplifying future studies to order $1/\Lambda^4$. The calculations include the decays of $W^\pm/Z$ to leptons to facilitate comparison with experimental LHC studies. By considering ratios of the integrated phase space and parton distribution function contributions to the SM, we were able to compare the phase space populations of various effective coupling combinations to those of the SM. Finally, motivated by these differences in phase space populations, we studied distributions for some of the more interesting coupling combinations. This serves to inform potential collider studies and experimental analyses, particularly in how to adapt searches that are SM inspired into searches optimized for the SMEFT, and which we hope may improve future fits to order $1/\Lambda^4$.

## Acknowledgments

The authors thank Michael Trott, Alessandro Broggio, and Simon Plätzer for useful discussions.

**Funding information** The work of A.M. was supported in part by the National Science Foundation under Grant Number PHY-2112540.

---

[10]We have chosen positive signs for simplicity, and – at least for the coefficients of dimension eight operators – motivated by analyticity constraints [46–48], though the bounds in the literature are more subtle than just requiring positive coefficients [49]. We have not attempted a detailed comparison here.

# A  Mapping between the effective Lagrangian and (geo)SMEFT

The effective couplings relevant to neutral current interactions can be written in terms of geometric invariants as follows:

$$c_{HZZ}^{(1)} = \frac{\bar{g}_Z^2}{2} \sqrt{h}^{44} \left[ \langle h_{33} \rangle \frac{v}{2} + \left\langle \frac{\delta h_{33}}{\delta h} \right\rangle \frac{v^2}{4} \right], \tag{A.1}$$

$$c_{HZZ}^{(2)} = -\frac{\sqrt{h}^{44}}{4} \bar{g}_Z^2 \left[ \left\langle \frac{\delta g_{33}}{\delta h} \right\rangle \frac{c_Z^4}{g_2^4} - 2 \left\langle \frac{\delta g_{34}}{\delta h} \right\rangle \frac{c_Z^2 s_Z^2}{g_1 g_2} + \left\langle \frac{\delta g_{44}}{\delta h} \right\rangle \frac{s_Z^4}{g_1^2} \right], \tag{A.2}$$

$$c_{HZZ}^{(3)} = \sqrt{h}^{44} \bar{g}_Z^2 v \left[ \langle k_{34}^3 \rangle \frac{c_Z^2}{g_2} - \langle k_{34}^4 \rangle \frac{s_Z^2}{g_1} \right], \tag{A.3}$$

$$c_{HAZ}^{(2)} = -\frac{\sqrt{h}^{44}}{2} \bar{e} \bar{g}_Z \left[ \left\langle \frac{\delta g_{33}}{\delta h} \right\rangle \frac{c_Z^2}{g_2^2} + \left\langle \frac{\delta g_{34}}{\delta h} \right\rangle \frac{c_Z^2 - s_Z^2}{g_1 g_2} - \left\langle \frac{\delta g_{44}}{\delta h} \right\rangle \frac{s_Z^2}{g_1^2} \right], \tag{A.4}$$

$$c_{HAZ}^{(3)} = -\sqrt{h}^{44} \bar{g}_Z \bar{e} v \left[ \langle k_{34}^3 \rangle \frac{1}{g_2} - \langle k_{34}^4 \rangle \frac{1}{g_1} \right], \tag{A.5}$$

$$g_{L,R,pr}^{Z\psi} = \frac{\bar{g}_Z}{2} \left( \left[ 2 s_Z^2 Q_\psi - (\sigma_3)_\psi \right] \delta_{pr} + (\sigma_3)_\psi v \langle L_{3,3}^{\psi,pr} \rangle + v \langle L_{3,4}^{\psi,pr} \rangle \right), \tag{A.6}$$

$$g_{L,R,pr}^{hZ\psi} = \frac{\bar{g}_Z}{2} \sqrt{h}^{44} \left\langle \left( v \frac{\delta}{\delta h} + 1 \right) \left[ (\sigma_3)_\psi L_{3,3}^{\psi,pr} + L_{3,4}^{\psi,pr} \right] \right\rangle, \tag{A.7}$$

$$g_{d,pr}^{Z\psi} = \frac{\bar{g}_Z v}{\sqrt{2}} \left[ \langle d_3^{\psi,pr} \rangle \frac{c_Z^2}{g_2} - \langle d_4^{\psi,pr} \rangle \frac{s_Z^2}{g_1} \right], \tag{A.8}$$

$$g_{d,pr}^{Zh\psi} = \frac{\bar{g}_Z \sqrt{h}^{44}}{\sqrt{2}} \left\langle \left( v \frac{\delta}{\delta h} + 1 \right) \left[ d_3^{\psi,pr} \frac{c_Z^2}{g_2} - d_4^{\psi,pr} \frac{s_Z^2}{g_1} \right] \right\rangle, \tag{A.9}$$

where $Q_\psi$ is the charge of a given chiral fermion $\psi$, and $(\sigma_3)_\psi$ is twice the third component of the isospin of $\psi$ (0 for right-handed fermions). The effective couplings relevant to charged current interactions can be written in terms of geometric invariants as follows:

$$c_{HWW}^{(1)} = \sqrt{h}^{44} \bar{g}_2^2 \left[ \left\langle \frac{\delta h_{11}}{\delta h} \right\rangle \frac{v^2}{4} + \langle h_{11} \rangle \frac{v}{2} \right], \tag{A.10}$$

$$c_{HWW}^{(2)} = -\sqrt{h}^{44} \langle g^{11} \rangle \left\langle \frac{\delta g_{11}}{\delta h} \right\rangle, \tag{A.11}$$

$$c_{HWW}^{(3)} = \bar{g}_2 \sqrt{h}^{44} \sqrt{g}^{11} v \left( \langle \kappa_{42}^2 \rangle - i \langle k_{42}^1 \rangle \right), \tag{A.12}$$

$$c_{HWW}^{(4)} = \bar{g}_2 \sqrt{h}^{44} \sqrt{g}^{11} v \left( \langle \kappa_{42}^2 \rangle + i \langle k_{42}^1 \rangle \right), \tag{A.13}$$

$$g_{L,pr}^{W\psi} = -\frac{\bar{g}_2}{\sqrt{2}} \left[ \delta_{pr} - v \langle L_{11}^{\psi,pr} \rangle + i v \langle L_{12}^{\psi,pr} \rangle \right], \tag{A.14}$$

$$g_{R,pr}^{W\psi} = \frac{\bar{g}_2 v}{\sqrt{2}} \langle L_1^{ud,pr} \rangle, \tag{A.15}$$

$$g_{L,pr}^{(1),hW\psi} = \frac{\bar{g}_2 \sqrt{h}^{44}}{\sqrt{2}} \left\langle \left( v \frac{\delta}{\delta h} + 1 \right) \left[ L_{11}^{\psi,pr} - i L_{12}^{\psi,pr} \right] \right\rangle, \tag{A.16}$$

$$g_{R,pr}^{(1),hW\psi} = \frac{\bar{g}_2 \sqrt{h}^{44}}{\sqrt{2}} \left\langle \left( v \frac{\delta}{\delta h} + 1 \right) L_1^{ud,pr} \right\rangle, \tag{A.17}$$

$$g_{d,pr}^{W\psi} = \frac{\sqrt{g}^{11}}{\sqrt{2}} \left( \langle d_1^{\psi,pr} \rangle + i \langle d_2^{\psi,pr} \rangle \right), \tag{A.18}$$

$$g_{d,pr}^{hW\psi} = \frac{\sqrt{g}^{11} \sqrt{h}^{44}}{2} \left\langle \left( v \frac{\delta}{\delta h} + 1 \right) \left( d_1^{\psi,pr} + i d_2^{\psi,pr} \right) \right\rangle. \tag{A.19}$$

The geometric quantities above, as well as $v$, $g_1$, and $g_2$ in terms of Wilson coefficients can be found in Appendix B.

The effective couplings of $hZ\bar{\psi}\psi$ can be expressed in terms of the Wilson coefficients as follows:[11]

$$-g_\pm^{(2),hZ\psi} = 0\,, \tag{A.20}$$

$$-g_\pm^{(3),hZ\psi} = 0\,, \tag{A.21}$$

$$-g_\pm^{(4),hZ\psi} = \bar{s}_W v\left(c_{\psi^2BH^2D}^{(1)} + (\sigma_3)_\psi c_{\psi^2BH^2D}^{(3)}\right)$$
$$+ \bar{c}_W v\left(c_{\psi^2WH^2D}^{(1)} - (\sigma_3)_\psi c_{\psi^2WH^2D}^{(3)}\right)$$
$$+ \frac{\bar{e}v}{2\bar{c}_W\bar{s}_W}\left(c_{\psi^2H^2D^3}^{(1)} - (\sigma_3)_\psi c_{\psi^2H^2D^3}^{(3)}\right)\,, \tag{A.22}$$

$$-g_\pm^{(5),hZ\psi} = -\bar{s}_W v\left(c_{\psi^2BH^2D}^{(1)} + (\sigma_3)_\psi c_{\psi^2BH^2D}^{(3)}\right)$$
$$- \bar{c}_W v\left(c_{\psi^2WH^2D}^{(1)} - (\sigma_3)_\psi c_{\psi^2WH^2D}^{(3)}\right)$$
$$+ \frac{\bar{e}v}{2\bar{c}_W\bar{s}_W}\left(c_{\psi^2H^2D^3}^{(1)} - (\sigma_3)_\psi c_{\psi^2H^2D^3}^{(3)}\right)\,, \tag{A.23}$$

$$-g_\pm^{(6),hZ\psi} = 0\,, \tag{A.24}$$

$$-g_\pm^{(7),hZ\psi} = -\frac{\bar{e}v}{\bar{c}_W\bar{s}_W}\left(c_{\psi^2H^2D^3}^{(1)} - (\sigma_3)_\psi c_{\psi^2H^2D^3}^{(3)}\right)\,, \tag{A.25}$$

$$-g_\pm^{(8),hZ\psi} = 0\,. \tag{A.26}$$

The effective couplings of $hW^\pm\bar{\psi}\psi$ can be expressed in terms of the Wilson coefficients as follows:

$$-g_L^{(2),hW^\pm\psi} = \pm\frac{\bar{e}v}{\sqrt{2}\bar{s}_W}c_{\psi^2H^2D^3}^{(4)}\,, \tag{A.27}$$

$$-g_L^{(3),hW^\pm\psi} = \mp\frac{\bar{e}v}{\sqrt{2}\bar{s}_W}c_{\psi^2H^2D^3}^{(4)}\,, \tag{A.28}$$

$$-g_L^{(4),hW^\pm\psi} = -\sqrt{2}v\left(c_{\psi^2WH^2D}^{(3)} \pm ic_{\psi^2WH^2D}^{(5)}\right)$$
$$- \frac{\bar{e}v}{\sqrt{2}\bar{s}_W}c_{\psi^2H^2D^3}^{(3)}\,, \tag{A.29}$$

$$-g_L^{(5),hW^\pm\psi} = \sqrt{2}v\left(c_{\psi^2WH^2D}^{(3)} \pm ic_{\psi^2WH^2D}^{(5)}\right)$$
$$- \frac{\bar{e}v}{\sqrt{2}\bar{s}_W}c_{\psi^2H^2D^3}^{(3)}\,, \tag{A.30}$$

$$-g_L^{(6),hW^\pm\psi} = \mp\frac{\bar{e}v}{\sqrt{2}\bar{s}_W}c_{\psi^2H^2D^3}^{(4)}\,, \tag{A.31}$$

$$-g_L^{(7),hW^\pm\psi} = \frac{\sqrt{2}\bar{e}v}{\bar{s}_W}c_{\psi^2H^2D^3}^{(3)}\,, \tag{A.32}$$

$$-g_L^{(8),hW^\pm\psi} = \pm\frac{\bar{e}v}{\sqrt{2}\bar{s}_W}c_{\psi^2H^2D^3}^{(4)}\,. \tag{A.33}$$

---

[11]Note: For all of the following, the subscript $\psi$ needs to be compared with the third column of Tabs. 1 and 2 which indicates for which fermions a given operator exists.

Table 7: The list of input parameters used in this article. The asterisk indicates that this value is derived from the others (i.e. it is not an input parameter). We further include the full widths as predicted by the SM and are necessary for the discussions of propagator corrections (see Sec. 2.3).

|  | $\{\hat{\alpha}, \hat{m}_Z, \hat{G}_F\}$ | $\{\hat{m}_W, \hat{m}_Z, \hat{G}_F\}$ |
|---|---|---|
| $\hat{\alpha}(\hat{m}_Z)$ | 0.00775 | 0.00756* |
| $\hat{m}_W$ | 79.9664 GeV* | 80.387 GeV |
| $\hat{G}_F$ | 1.1663787·$10^{-5}$ $[\text{GeV}]^{-2}$ | |
| $\hat{m}_Z$ | 91.1876 GeV | |
| $\hat{m}_H$ | 125.1 GeV | |
| $\Gamma_Z$ | 2.42 GeV | 2.44 GeV |
| $\Gamma_W$ | 2.01 GeV | 2.05 GeV |

# B    geoSMEFT conventions

## B.1    Input parameters and SM-like couplings

We begin this discussion with the couplings like those occurring in the renormalizable Lagrangian and their relation to the two input parameter schemes. Our input parameters are taken from [14] and reproduced here in Tab. 7.

We will define the relation between the true vacuum expectation value and $\hat{v}$ as:

$$v^2 \equiv \hat{v}^2 + \hat{v}^4 \delta G_F^{(6)} + \hat{v}^6 \delta G_F^{(8)} - \frac{\hat{v}^6}{4} \left( c_{HD}^{(8)} - c_{HD,2}^{(8)} \right). \tag{B.1}$$

Note this definition is potentially problematic as we have implicitly included terms such as the square of $\delta G_F^{(6)}$ in the definition of $\delta G_F^{(8)}$. This is for convenience as our calculations are only sensitive to the sum of the two terms. We have also explicitly written the dependence of $c_{HD}^{(8)}$ and $c_{HD,2}^{(8)}$, this is because our calculations are sensitive to these Wilson coefficients from other aspects of the calculations.[12]

Combining this with Appendix D of [23] we are able to formulate Table 8. $\bar{g}_2$ can be derived simply from $g_2$ given that $\bar{g}_2 = \sqrt{g}^{11} g_2$ and Eq. B.10 in the next subsection. From Tab. 8 we can identify the corrections to $\bar{m}_W$ with the $\delta m_W$ of Eq. 15:

$$\bar{m}_W \equiv m_W^{(\text{SM})} + \delta m_W \sim 80 + \underbrace{\left( -63 c_{HWB}^{(6)} \hat{v}^2 - 29 \tilde{c}_{HD}^{(6)} \hat{v}^2 + \cdots \right)}_{\delta m_W}. \tag{B.2}$$

Notice that we include both the $1/\Lambda^2$ and $1/\Lambda^4$ dependence in $\delta m_W$ so it is implicit we need to truncate at the appropriate order.

## B.2    Expectations of field-space connections

Here we give the relation between the relevant expectation values of field-space connections and the Wilson coefficients of the SMEFT. They are derived using the definitions in Eqs. 5 through 9. The terms from the $h$ and $g$ field-space connections appear frequently as they shift many effective couplings:

---

[12]Indeed if we did not separate this part out, we would incorrectly find the quantities in Tab. 8 depend on $c_{HD}^{(8)}$. The dependence cancels between writing $v$ as in Eq. B.1 and the other $c_{HD}^{(8)}$ dependence of the quantities.

Table 8: $\bar{c}_W$ and $\bar{c}_Z$ can be derives from $\bar{s}_W$ and $\bar{s}_Z$ using the usual trigonometric identity $c = \sqrt{1-s^2}$. Also note $\bar{g}_Z$ is the same for both sets of input parameters. $\bar{e} = \hat{e}$ receives no corrections in the in the $\alpha$ scheme, just as $\bar{m}_W = \hat{m}_W$ receives no corrections in the $m_W$ scheme.

| | $\alpha$ scheme | | | | | | $m_W$ scheme | | | | |
|---|---|---|---|---|---|---|---|---|---|---|---|
| | $\bar{s}_W$ | $\bar{s}_Z$ | $g_1$ | $g_2$ | $\bar{g}_Z$ | $\bar{m}_W$ | $\bar{s}_W$ | $\bar{s}_Z$ | $g_1$ | $g_2$ | $\bar{e}$ |
| const. | 0.48 | 0.48 | 0.36 | 0.65 | 0.74 | 80 | 0.47 | 0.47 | 0.35 | 0.65 | 0.31 |
| $\tilde{c}_{HB}^{(6)}$ | 0 | 0 | -0.36 | 0 | 0 | 0 | 0 | 0 | -0.35 | 0 | 0 |
| $\tilde{c}_{HW}^{(6)}$ | 0 | 0 | 0 | -0.65 | 0 | 0 | -0.82 | -0.94 | -1.3 | -0.33 | -0.54 |
| $\tilde{c}_{HWB}^{(6)}$ | 0.81 | 0.81 | 0.28 | -0.51 | 0 | -63 | -0.44 | 0 | -0.65 | 0 | 0 |
| $\tilde{c}_{HD}^{(6)}$ | 0.17 | 0.17 | 0.038 | -0.23 | -0.19 | -29 | -0.41 | -0.41 | -0.39 | 0 | -0.27 |
| $\tilde{\delta G}_F^{(6)}$ | 0.34 | 0.34 | 0.076 | -0.46 | -0.37 | -17 | 0 | 0 | -0.17 | -0.33 | -0.15 |
| $\tilde{c}_{HB}^{(8)}$ | 0 | 0 | -0.18 | 0 | 0 | 0 | 0 | 0 | -0.17 | 0 | 0 |
| $\tilde{c}_{HW}^{(8)}$ | 0 | 0 | 0 | -0.32 | 0 | 0 | -0.41 | -0.26 | -0.65 | -0.16 | 0 |
| $\tilde{c}_{HWB}^{(8)}$ | 0.41 | 0.41 | 0.14 | -0.25 | 0 | -31 | -0.22 | 0.11 | -0.33 | 0 | 0.14 |
| $\tilde{c}_{HW,2}^{(8)}$ | 0 | 0 | 0 | -0.32 | 0 | -40 | -0.82 | -0.41 | -0.96 | 0 | 0.15 |
| $c_{HD,2}^{(8)}$ | 0.17 | 0.17 | 0.038 | -0.23 | -0.19 | -29 | -0.41 | -0.21 | -0.39 | 0 | 0 |
| $\delta G_F^{(8)}$ | 0.34 | 0.34 | 0.076 | -0.46 | -0.37 | -17 | 0 | 0 | -0.17 | -0.33 | -0.15 |
| $\left[\tilde{c}_{HB}^{(6)}\right]^2$ | 0 | 0 | -0.18 | 0 | 0 | 0 | 0 | 0 | -0.17 | 0 | 0 |
| $\tilde{c}_{HB}^{(6)}\tilde{c}_{HW}^{(6)}$ | 0 | 0 | 0 | 0 | 0 | 0 | 0 | 0 | 1.3 | 0 | 0 |
| $\tilde{c}_{HB}^{(6)}\tilde{c}_{HWB}^{(6)}$ | 1.0 | 0.81 | 0 | -0.51 | 0 | -63 | 1.1 | 0.96 | 1.3 | 0 | -0.14 |
| $\tilde{c}_{HB}^{(6)}\tilde{c}_{HD}^{(6)}$ | 0 | 0 | -0.038 | 0 | 0 | 0 | 0 | 0 | 0.39 | 0 | 0 |
| $\tilde{c}_{HB}^{(6)}\delta\tilde{G}_F^{(6)}$ | 0 | 0 | -0.43 | 0 | 0 | 0 | 0 | 0 | -0.17 | 0 | 0 |
| $\left[\tilde{c}_{HW}^{(6)}\right]^2$ | 0 | 0 | 0 | -0.32 | 0 | 0 | -2.0 | -1.8 | -0.23 | -0.24 | 0 |
| $\tilde{c}_{HW}^{(6)}\tilde{c}_{HWB}^{(6)}$ | 0.60 | 0.81 | 0.28 | 0 | 0 | -63 | -0.44 | -0.82 | 0.094 | 0 | -0.14 |
| $\tilde{c}_{HW}^{(6)}\tilde{c}_{HD}^{(6)}$ | 0 | 0 | 0 | 0.23 | 0 | 0 | -1.1 | -1.4 | 0.10 | 0 | 0 |
| $\tilde{c}_{HW}^{(6)}\delta\tilde{G}_F^{(6)}$ | 0 | 0 | 0 | -0.19 | 0 | 0 | -0.82 | -0.74 | -0.65 | -0.16 | 0 |
| $\left[\tilde{c}_{HWB}^{(6)}\right]^2$ | 0.64 | 0.50 | 0.59 | -1.0 | 0 | -120 | -0.059 | -0.18 | -0.35 | 0 | 0.12 |
| $\tilde{c}_{HWB}^{(6)}\tilde{c}_{HD}^{(6)}$ | 0.41 | 0.41 | 0.27 | -0.26 | 0 | -32 | -0.11 | -0.49 | 0 | 0 | 0 |
| $\tilde{c}_{HWB}^{(6)}\delta\tilde{G}_F^{(6)}$ | 1.6 | 1.6 | 0.82 | -1.0 | 0 | -160 | -0.44 | 0.11 | -0.33 | 0 | 0.14 |
| $\left[\tilde{c}_{HD}^{(6)}\right]^2$ | 0.022 | 0.022 | 0.018 | 0.053 | 0.069 | 6.5 | -0.18 | -0.32 | -0.024 | 0 | 0 |
| $\tilde{c}_{HD}^{(6)}\delta\tilde{G}_F^{(6)}$ | 0.43 | 0.43 | 0.15 | -0.25 | -0.093 | -45 | -0.41 | -0.31 | -0.20 | 0 | 0 |
| $\left[\delta\tilde{G}_F\right]^2$ | 0.088 | 0.088 | 0.071 | 0.21 | 0.28 | -12 | 0 | 0 | 0.13 | 0.24 | 0.12 |

$$\langle h_{11} \rangle = 1 + \frac{1}{4}\left(c_{HD}^{(8)} - c_{HD,2}^{(8)}\right)v^4, \tag{B.3}$$

$$\langle h_{33} \rangle = 1 + \frac{1}{2}c_{HD}^{(6)}v^2 + \frac{1}{4}\left(c_{HD}^{(8)} + c_{HD,2}^{(8)}\right)v^4, \tag{B.4}$$

$$\langle h^{44} \rangle = 1 + \frac{1}{2}\left(4c_{H\square}^{(6)} - c_{HD}^{(6)}\right) - \frac{1}{4}\left[c_{HD}^{(8)} + c_{HD,2}^{(8)} - \left(c_{HD}^{(6)} - 4c_{H\square}^{(6)}\right)^2\right]v^4, \tag{B.5}$$

$$\sqrt{h}^{44} = 1 + \frac{v^2}{4}\left(4c_{H\square}^{(6)} - c_{HD}^{(6)}\right) + \frac{v^4}{32}\left[3\left(c_{HD}^{(6)} - 4c_{H\square}^{(6)}\right)^2 - 4c_{HD}^{(8)} - 4c_{HD,2}^{(8)}\right], \tag{B.6}$$

$$\left\langle \frac{\delta h_{11}}{\delta h} \right\rangle = \left(c_{HD}^{(8)} - c_{HD,2}^{(8)}\right)v^3, \tag{B.7}$$

$$\left\langle \frac{\delta h_{33}}{\delta h} \right\rangle = c_{HD}^{(6)}v + \frac{1}{4}\left[4c_{HD}^{(8)} + 4c_{HD,2}^{(8)} + \left(4c_{H\square}^{(6)} - c_{HD}^{(6)}\right)c_{HD}^{(6)}\right]v^3, \tag{B.8}$$

$$\left\langle g^{11} \right\rangle = 1 + 2c_{HW}^{(6)}v^2 + \left[4\left(c_{HW}^{(6)}\right)^2 + c_{HW}^{(8)}\right]v^4, \tag{B.9}$$

$$\sqrt{g}^{11} = 1 + c_{HW}^{(6)}v^2 + \frac{v^4}{2}\left(3\left[c_{HW}^{(6)}\right]^2 + c_{HW}^{(8)}\right), \tag{B.10}$$

$$\left\langle \frac{\delta g_{11}}{\delta h} \right\rangle = -4c_{HW}^{(6)}v - \left[4c_{HW}^{(8)} + 4c_{HW}^{(6)}c_{H\square}^{(6)} - c_{HW}^{(6)}c_{HD}^{(6)}\right]v^3, \tag{B.11}$$

$$\left\langle \frac{\delta g_{33}}{\delta h} \right\rangle = \left\langle \frac{\delta g_{11}}{\delta h} \right\rangle - 4c_{HW,2}^{(8)}v^3, \tag{B.12}$$

$$\left\langle \frac{\delta g_{34}}{\delta h} \right\rangle = 2c_{HWB}^{(6)}v + \frac{1}{2}\left[4c_{HWB}^{(8)} + 4c_{H\square}^{(6)}c_{HWB}^{(6)} - c_{HWB}^{(6)}c_{HD}^{(6)}\right]v^3, \tag{B.13}$$

$$\left\langle \frac{\delta g_{44}}{\delta h} \right\rangle = -4c_{HB}^{(6)}v - \left[4c_{HB}^{(8)} + c_{HB}^{(6)}c_{H\square}^{(6)} - c_{HB}^{(6)}c_{HD}^{(6)}\right]v^3. \tag{B.14}$$

The field-space connections $\kappa_{IJ}^A$ give rise to new $HVV$ interactions that are not present in the SM at tree level. The relevant combinations for our analysis are:

$$\left\langle \kappa_{34}^3 \right\rangle = \frac{1}{2}c_{HDHW}^{(6)} + \frac{1}{4}c_{HDHW}^{(8)}v^2, \tag{B.15}$$

$$\left\langle \kappa_{34}^4 \right\rangle = -\frac{1}{2}c_{HDHB}^{(6)} - \frac{1}{4}c_{HDHB}^{(8)}v^2, \tag{B.16}$$

$$\left\langle \kappa_{42}^1 \right\rangle = -\frac{1}{4}c_{HDHW,2}^{(8)}v^2, \tag{B.17}$$

$$\left\langle \kappa_{42}^2 \right\rangle = -\frac{1}{2}c_{HDHW}^{(6)} - \frac{1}{4}c_{HDHW}^{(8)}v^2. \tag{B.18}$$

Terms coming from $L$ shift $V\bar{\psi}\psi$ and $Vh\bar{\psi}\psi$ vertices. They have implicit dependence on $\langle h_{IJ} \rangle$ as they depend on the scalar fields. We have only written the expressions for the right-handed quarks, $u$ and $d$, and the left-handed quark doublet $q$. The leptonic equivalents can be obtain for the right-handed charged leptons by taking the $d \to e$ and for the left-handed leptons by $q \to \ell$.

$$\left\langle L_{33}^u \right\rangle = \left\langle L_{33}^d \right\rangle = 0, \tag{B.19}$$

$$\left\langle L_{34}^u \right\rangle = c_{Hu}^{(6)}v + \frac{1}{2}c_{Hu}^{(8)}v^3, \tag{B.20}$$

$$\left\langle L_{34}^d \right\rangle = c_{Hd}^{(6)}v + \frac{1}{2}c_{Hd}^{(8)}v^3, \tag{B.21}$$

$$\left\langle L_{33}^q \right\rangle = -c_{Hq}^{3,(6)}v - \frac{1}{2}\left[c_{Hq}^{2,(8)} + c_{Hq}^{3,(8)}\right]v^3, \tag{B.22}$$

$$\left\langle L_{34}^q \right\rangle = c_{Hq}^{1,(6)}v + \frac{1}{2}c_{Hq}^{1,(8)}v^3, \tag{B.23}$$

$$\left\langle \frac{\delta L_{34}^u}{\delta h} \right\rangle = c_{Hu}^{(6)} + \frac{1}{4}\left[ 6c_{Hu}^{(8)} + \left( 4c_{H\square}^{(6)} - c_{HD}^{(6)} \right) c_{Hu}^{(6)} \right] v^2 \,, \tag{B.24}$$

$$\left\langle \frac{\delta L_{34}^d}{\delta h} \right\rangle = c_{Hd}^{(6)} + \frac{1}{4}\left[ 6c_{Hd}^{(8)} + \left( 4c_{H\square}^{(6)} - c_{HD}^{(6)} \right) c_{Hd}^{(6)} \right] v^2 \,, \tag{B.25}$$

$$\left\langle \frac{\delta L_{33}^q}{\delta h} \right\rangle = -c_{Hq}^{3,(6)} - \frac{1}{4}\left[ 6c_{Hq}^{2,(8)} + 6c_{Hq}^{3,(8)} + \left( 4c_{H\square}^{(6)} - c_{HD}^{(6)} \right) c_{Hq}^{3,(8)} \right] v^2 \,, \tag{B.26}$$

$$\left\langle \frac{\delta L_{34}^q}{\delta h} \right\rangle = c_{Hq}^{1,(6)} + \frac{1}{4}\left[ 6c_{Hq}^{1,(8)} + \left( 4c_{H\square}^{(6)} - c_{HD}^{(6)} \right) c_{Hq}^{1,(6)} \right] v^2 \,, \tag{B.27}$$

$$\left\langle L_{11}^q \right\rangle = -c_{Hq}^{3,(6)} v - \frac{1}{2}c_{Hq}^{3,(6)} v^3 \,, \tag{B.28}$$

$$\left\langle L_{12}^q \right\rangle = -\frac{1}{2}c_{Hq}^{\epsilon(8)} v^3 \,, \tag{B.29}$$

$$\left\langle \frac{\delta L_{11}^q}{\delta h} \right\rangle = -c_{Hq}^{3,(6)} - \frac{1}{4}\left[ 6c_{Hq}^{3,(8)} + \left( 4c_{H\square}^{(6)} - c_{HD}^{(6)} \right) c_{Hq}^{3,(6)} \right] v^2 \,, \tag{B.30}$$

$$\left\langle \frac{\delta L_{12}^q}{\delta h} \right\rangle = -\frac{3}{2}c_{Hq}^{\epsilon(8)} v^2 \,, \tag{B.31}$$

$$\left\langle L_1^{ud} \right\rangle = \frac{1}{2}c_{Hud}^{(6)} v + \frac{1}{4}c_{Hud}^{(8)} v^3 \,, \tag{B.32}$$

$$\left\langle \frac{\delta L_1^{ud}}{\delta h} \right\rangle = \frac{1}{2}c_{Hud}^{(6)} + \frac{1}{8}\left[ 6c_{Hud}^{(8)} + c_{Hud}^{(6)} \left( 4c_{H\square}^{(6)} - c_{HD}^{(6)} \right) \right] v^2 \,. \tag{B.33}$$

With all of the above it is important to note that $v \neq \hat{v}$, and therefore the relation between the vacuum expectation value and the input value $\hat{v}$ need to be added.

## C Phase space integration

Our short review of the phase space integration primarily follows [50]. The phase space integration for the scattering $p_1, p_2 \rightarrow p_3, p_4, p_5$ can be written as:

$$d\Pi_{\text{LIPS}} = \frac{\pi}{16s_{12}} \frac{ds_{14} ds_{15} ds_{35} ds_{45}}{\sqrt{-\Delta_4}} \,, \tag{C.1}$$

where $s_{ij} = (p_i + p_j)^2$ and $\Delta_4$ is the symmetric Gram determinant of any four of the momenta, for example:

$$\det|G_4(p_1, p_2, p_3, p_4)| = \begin{vmatrix} p_1^2 & p_1 \cdot p_2 & p_1 \cdot p_3 & p_1 \cdot p_4 \\ p_1 \cdot p_2 & p_2^2 & p_2 \cdot p_3 & p_2 \cdot p_4 \\ p_1 \cdot p_3 & p_2 \cdot p_3 & p_3^2 & p_3 \cdot p_4 \\ p_1 \cdot p_4 & p_2 \cdot p_4 & p_3 \cdot p_4 & p_4^2 \end{vmatrix} \,. \tag{C.2}$$

The physical region of integration is given by the conditions: $\Delta_1 < 0$, $\Delta_2 < 0$, $\Delta_3 < 0$, and $\Delta_4 < 0$ where $\Delta_i$ is the coefficient of $\lambda^i$ in,

$$\det|\lambda \mathbf{1}_{4\times4} - G_4(p_a, p_b, p_c, p_d)| \,. \tag{C.3}$$

The partonic cross section is then given by:

$$d\hat{\sigma} = \frac{1}{2s_{12}} \frac{1}{3} \frac{1}{4} \frac{1}{(2\pi)^5} |\mathcal{M}_i|^2 d\Pi_{\text{LIPS}} \,. \tag{C.4}$$

Note this definition includes symmetry factors specific to the process $\bar{q}q \rightarrow h\bar{l}l$ such as color and spin averaging factors. Further, $|\mathcal{M}_i|^2$ holds the place for not only squares of amplitudes, but also their interference, meaning an individual $\hat{\sigma}_i$ is not necessarily positive-definite.

The parton distribution functions are folded in as:

$$d\sigma_i = d\hat{\sigma}_i \left[ x_1 f_q(x_1) x_2 f_{\bar{q}}(x_2) + x_2 f_1(x_2) x_1 f_{\bar{q}}(x_1) \right] \frac{ds_{12} dY}{s_{12}}, \qquad \text{(C.5)}$$

where $Y$ is the boost rapidity. With this definition we have that:

$$x_1 = \sqrt{\frac{s_{12}}{s_{\text{LHC}}}} e^Y, \qquad \text{(C.6)}$$

$$x_2 = \sqrt{\frac{s_{12}}{s_{\text{LHC}}}} e^{-Y}. \qquad \text{(C.7)}$$

With $S_{\text{LHC}}$ is the LHC center of mass energy, taken to be 13 TeV in this article. The regions of integration are then given by the solution to $\Delta_1 < 0$, $\Delta_2 < 0$, $\Delta_3 < 0$, and $\Delta_4 < 0$ combined with:

$$\begin{aligned} m_H^2 &\leq s_{12} \leq S_{\text{LHC}}, \\ \log\left(\sqrt{\frac{s_{12}}{S_{\text{LHC}}}}\right) &\leq Y \leq -\log\left(\sqrt{\frac{s_{12}}{S_{\text{LHC}}}}\right). \end{aligned} \qquad \text{(C.8)}$$

We do not have an exact solution to $\Delta_1 < 0$, $\Delta_2 < 0$, $\Delta_3 < 0$, and $\Delta_4 < 0$, however it can be shown that:

$$\begin{aligned} 0 &< s_{45} < s_{12} + \bar{m}_H^2 - 2\sqrt{s_{12}\bar{m}_H^2}, \\ \tfrac{1}{2}(s_{12} + \bar{m}_H^2 - s_{45}) - \kappa &< s_{35} < \tfrac{1}{2}(s_{12} + \bar{m}_H^2 - s_{45}) + \kappa, \\ 0 &< s_{14} < s_{12} - s_{35}, \\ 0 &< s_{15} < \frac{(s_{12} - s_{14})(s_{35} - \bar{m}_H^2)}{s_{35}}, \end{aligned} \qquad \text{(C.9)}$$

for massless fermions, and we have used:

$$\kappa = \frac{1}{2}\sqrt{s_{12}^2 + s_{45}^2 + \bar{m}_H^4 - 2\bar{m}_H^2 s_{45} - 2\bar{m}_H^2 s_{12} - 2s_{12}s_{45}}. \qquad \text{(C.10)}$$

We stress again, *these are not the exact integration limits*. These limits include all physical points as well as some nonphysical points in the phase space. We still require $\Delta_1 < 0$, $\Delta_2 < 0$, $\Delta_3 < 0$, and $\Delta_4 < 0$ of our integration routine, which removes the nonphysical points. It is also useful to carefully select how each $s_{ij}$ is sampled. Particularly $s_{45}$, which corresponds to the invariant mass of the final-state fermion pair, should be sampled according to a Breit-Wigner distribution.

# D  Converting $D^3\psi^2 H^2$ operators to geoSMEFT compliant basis form

To form a geoSMEFT compliant choice, one can use the algorithm in Ref. [26] to shuffle derivatives. However, rather than massaging existing bases, it is often faster to construct a new, compliant basis from scratch using the techniques in Ref. [51]. Following the procedure in Ref. [51], operators in the $D^3\psi^2 H^2$ class are formed by all spinor contractions of

$$\tilde{\lambda}_1 \lambda_2 (\tilde{\lambda}_i \lambda_i)(\tilde{\lambda}_j \lambda_j)(\tilde{\lambda}_k \lambda_k), \qquad \text{(D.1)}$$

after accounting for EOM, IBP, and usual spinor tricks. Here particle 1 and 2 are $\bar{\psi}$ and $\psi$ (with momenta $p_1$, $p_2$), assumed for simplicity to be EW singlets, $H^\dagger$ and $H$ are particles 3 and 4; $i, j, k$ are momenta, free to take values $p_1$ through $p_4$ but subject to EOM, in the form $p_i^2 = 0$ and IBP, in the form $p_1 + p_2 + p_3 + p_4 = 0$. Exploiting the Schouten identity, all products can be reduced to

$$[1i]\langle 2i\rangle s_{jk}, \tag{D.2}$$

where $s_{jk} = p_j \cdot p_k$. Spinor antisymmetry implies $i \neq p_1, p_2$, so $p_3, p_4$ are the only choices. These are related by total momentum conservation, so only one is independent. Without any loss of generality, let's choose $i = p_3$. With four fields present in the operator, the factor $s_{jk}$ is one of the Mandelstam variables, $s, t, u$. Of these, one can be removed by $s + t + u = 0$, so we are left with two independent operators, e.g.:

$$[13]\langle 23\rangle s, [13]\langle 23\rangle t. \tag{D.3}$$

Now we work in geoSMEFT compliance. This amounts to choosing the momenta in $s$ and $t$ – derivatives once translated back to operator form – to include as Higgs momenta $p_3, p_4$ whenever possible, i.e.

$$[13]\langle 23\rangle s_{34}, [13]\langle 23\rangle s_{24}. \tag{D.4}$$

Rewritten as operators, these are:

$$(\psi^\dagger \bar{\sigma}^\mu \psi)(D_{(\mu,} D_{\nu)} H^\dagger)(D_\nu H), \quad (\psi^\dagger \bar{\sigma}^\mu D_\nu \psi)(D_\mu H^\dagger)(D_\nu H), \tag{D.5}$$

clearly geoSMEFT compliant as expected. As shown, these are not hermitian, so we must add $+h.c.$. Exchanging $\psi$ for an electroweak doublet doubles the number of operators as two EW contractions are then possible, but each EW contraction can be massaged as above into geoSMEFT compliant form.

# E  Additional tables

Table 9: Same as Tab. 6, but using $\hat{m}_W$ scheme.

| $(\mathcal{L}_{\text{eff}}$ dependence$)/N_c$ | \multicolumn{4}{c|}{partons} | | | |
| --- | --- | --- | --- | --- |
| | $u\bar{d}$ | $c\bar{s}$ | $\bar{u}d$ | $\bar{c}s$ |
| $[c_{HWW}^{(1)}]^2 \lvert g_L^{W\ell}\rvert^2 \lvert g_L^{Wq}\rvert^2 \cdot 10^{-4}$ | 5.60 | 0.56 | 3.28 | 0.62 |
| $[c_{HWW}^{(2)}]^2 \lvert g_L^{W\ell}\rvert^2 \lvert g_L^{Wq}\rvert^2 \hat{v}^4$ | 0.30 | 0.22 | 0.28 | 0.23 |
| $c_{HWW}^{(1)} c_{HWW}^{(2)} \lvert g_L^{W\ell}\rvert^2 \lvert g_L^{Wq}\rvert^2 \hat{v}^2$ | 0.79 | 0.75 | 0.78 | 0.76 |
| $c_{HWW}^{(1)} c_{HWW}^{(3)} \lvert g_L^{W\ell}\rvert^2 \lvert g_L^{Wq}\rvert^2 \hat{v}^2$ | 0.58 | 0.54 | -2.67 | -2.12 |
| $c_{HWW}^{(1)} c_{HWW}^{(4)} \lvert g_L^{W\ell}\rvert^2 \lvert g_L^{Wq}\rvert^2 \hat{v}^2$ | -2.9 | -2.1 | 0.57 | 0.54 |
| $[c_{HWW}^{(1)}]^2 \lvert g_L^{W\ell}\rvert^2 \lvert g_R^{Wq}\rvert^2 \cdot 10^{-4}$ | 5.60 | 0.56 | 3.28 | 0.62 |
| $\lvert g_L^{W\ell}\rvert^2 \lvert g_L^{(1),hWq}\rvert^2 \hat{v}^4$ | 10.79 | 3.48 | 8.17 | 4.12 |
| $c_{HWW}^{(1)} \lvert g_L^{W\ell}\rvert^2 \text{Re}[(g_L^{(1),hWq})^* g_L^{Wq}] \hat{v}^2$ | 1.74 | 1.33 | 1.62 | 1.37 |
| $c_{HWW}^{(2)} \lvert g_L^{W\ell}\rvert^2 \text{Re}[(g_L^{(1),hWq})^* g_L^{Wq}] \hat{v}^2$ | 1.66 | 1.14 | 1.51 | 1.19 |
| $c_{HWW}^{(1)} \lvert g_L^{W\ell}\rvert^2 \text{Re}[(g_L^{(4),hWq})^* g_L^{Wq}] \hat{v}^4$ | 5.17 | 1.57 | 3.87 | 1.88 |
| $c_{HWW}^{(1)} \lvert g_L^{W\ell}\rvert^2 \text{Re}[(g_L^{(5),hWq})^* g_L^{Wq}] \hat{v}^4$ | 4.53 | 1.14 | 3.30 | 1.43 |
| $c_{HWW}^{(1)} \lvert g_L^{W\ell}\rvert^2 \text{Im}[(g_L^{(1),hWq})^* g_L^{Wq}] \hat{v}^2 \cdot 10^{-3}$ | -2.72* | -2.72* | -2.72* | -2.72* |
| $c_{HWW}^{(1)} \lvert g_L^{W\ell}\rvert^2 \text{Im}[(g_L^{(4),hWq})^* g_L^{Wq}] \hat{v}^4 \cdot 10^{-3}$ | -2.01* | -1.46* | -1.85* | -1.51* |
| $c_{HWW}^{(1)} \lvert g_L^{W\ell}\rvert^2 \text{Im}[(g_L^{(5),hWq})^* g_L^{Wq}] \hat{v}^4 \cdot 10^{-3}$ | -1.22* | -0.72* | -1.07* | -0.76* |
| $c_{HWW}^{(1)} \lvert g_L^{W\ell}\rvert^2 g_L^{(2),hWq} g_L^{Wq} \hat{v}^4$ | -2.81 | -0.95 | -2.15 | -1.12 |
| $c_{HWW}^{(1)} \lvert g_L^{W\ell}\rvert^2 g_L^{(3),hWq} g_L^{Wq} \hat{v}^4$ | -2.81 | -0.95 | -2.15 | -1.12 |
| $c_{HWW}^{(1)} \lvert g_L^{W\ell}\rvert^2 g_L^{(6),hWq} g_L^{Wq} \hat{v}^4$ | 2.27 | 0.57 | 1.65 | 0.72 |
| $c_{HWW}^{(1)} \lvert g_L^{W\ell}\rvert^2 g_L^{(7),hWq} g_L^{Wq} \hat{v}^4$ | -4.53 | -1.14 | -3.30 | -1.43 |
| $c_{HWW}^{(1)} \lvert g_L^{W\ell}\rvert^2 g_L^{(8),hWq} g_L^{Wq} \hat{v}^4$ | 2.27 | 0.57 | 1.65 | 0.72 |
| $[c_{HWW}^{(1)}]^2 \lvert g_L^{W\ell}\rvert^2 \lvert g_L^{Wq}\rvert^2 \delta\Gamma$ | -0.49 | -0.49 | -0.49 | -0.49 |
| $[c_{HWW}^{(1)}]^2 \lvert g_L^{W\ell}\rvert^2 \lvert g_L^{Wq}\rvert^2 \delta\Gamma^2$ | 0.24 | 0.24 | 0.24 | 0.24 |
| $c_{HWW}^{(1)} c_{HWW}^{(2)} \lvert g_L^{W\ell}\rvert^2 \lvert g_L^{Wq}\rvert^2 \delta\Gamma \hat{v}^2$ | -0.39 | -0.37 | -0.38 | -0.37 |
| $c_{HWW}^{(1)} \lvert g_L^{W\ell}\rvert^2 \text{Re}[(g_L^{(1),hWq})^*] g_L^{Wq} \delta\Gamma \hat{v}^2$ | -0.86 | -0.66 | -0.80 | -0.67 |

Table 10: Same as Tab. 4, but with the restriction $\hat{s} > (500 \text{ GeV})^2$.

| ($\mathcal{L}_{\text{eff}}$ dependence)/$N_c$ | partons | | | | |
|---|---|---|---|---|---|
| | dd | uu | ss | cc | bb |
| $[c_{HZZ}^{(1)}]^2 [g_L^{Z\ell}]^2 [g_L^{Zq}]^2 \cdot 10^{-4}$ | 0.271 | 0.135 | 0.0149 | 0.0120 | 0.0032 |
| $c_{HZZ}^{(1)} c_{HZZ}^{(2)} [g_L^{Z\ell}]^2 [g_L^{Zq}]^2 \hat{v}^2$ | 1.5 | 1.5 | 1.4 | 1.4 | 1.4 |
| $c_{HZZ}^{(1)} c_{HZZ}^{(3)} [g_L^{Z\ell}]^2 [g_L^{Zq}]^2 \hat{v}^2$ | 14 | 13 | 12 | 11 | 11 |
| $c_{HAZ}^{(2)} c_{HZZ}^{(1)} \bar{e} [g_L^{Z\ell}]^2 g_L^{Zq} Q_q \hat{v}^2$ | -1.4 | -1.4 | -1.4 | -1.4 | -1.4 |
| $c_{HAZ}^{(3)} c_{HZZ}^{(1)} \bar{e} [g_L^{Z\ell}]^2 g_L^{Zq} Q_q \hat{v}^2$ | -15 | -14 | -12 | -12 | -12 |
| $[c_{HZZ}^{(2)}]^2 [g_L^{Z\ell}]^2 [g_L^{Zq}]^2 \hat{v}^4$ | 2.2 | 2.1 | 1.8 | 1.7 | 1.7 |
| $[c_{HAZ}^{(2)}]^2 \bar{e}^2 [g_L^{Z\ell}]^2 Q_q^2 \hat{v}^4$ | 4.3 | 4.0 | 3.5 | 3.3 | 3.2 |
| $c_{HAZ}^{(2)} c_{HZZ}^{(2)} \bar{e} [g_L^{Z\ell}]^2 g_L^{Zq} Q_q \hat{v}^4$ | -4.4 | -4.1 | -3.6 | -3.4 | -3.3 |
| $c_{HZZ}^{(1)} [g_L^{Z\ell}]^2 g_L^{Zq} g_L^{(1),hZq} \hat{v}^2$ | 17 | 16 | 14 | 13 | 13 |
| $[g_L^{Z\ell}]^2 [g_L^{(1),hZq}]^2 \hat{v}^4$ | 140 | 110 | 75 | 62 | 59 |
| $c_{HZZ}^{(1)} [g_L^{Z\ell}]^2 g_L^{Zq} g_L^{(4),hZq} \hat{v}^4$ | 130 | 110 | 73 | 61 | 57 |
| $c_{HZZ}^{(1)} [g_L^{Z\ell}]^2 g_L^{Zq} g_L^{(5),hZq} \hat{v}^4$ | 120 | 100 | 65 | 53 | 49 |
| $c_{HZZ}^{(1)} [g_L^{Z\ell}]^2 g_L^{Zq} g_L^{(7),hZq} \hat{v}^4$ | -120 | -100 | -65 | -53 | -49 |
| $c_{HAZ}^{(2)} \bar{e} [g_L^{Z\ell}]^2 g_L^{(1),hZq} Q_q \hat{v}^4$ | -12 | -12 | -10 | -10 | -9 |
| $c_{HZZ}^{(2)} [g_L^{Z\ell}]^2 g_L^{Zq} g_L^{(1),hZq} \hat{v}^4$ | 13 | 12 | 10 | 9.8 | 9.6 |
| $[c_{HZZ}^{(1)}]^2 [g_L^{Z\ell}]^2 [g_L^{Zq}]^2 \delta\Gamma$ | -0.41 | -0.41 | -0.41 | 0.41 | -0.41 |
| $[c_{HZZ}^{(1)}]^2 [g_L^{Z\ell}]^2 [g_L^{Zq}]^2 \delta\Gamma^2$ | 0.17 | 0.17 | 0.17 | 0.17 | 0.17 |
| $c_{HZZ}^{(1)} c_{HZZ}^{(2)} [g_L^{Z\ell}]^2 [g_L^{Zq}]^2 \delta\Gamma \hat{v}^2$ | -0.59 | -0.59 | -0.58 | -0.58 | -0.58 |
| $c_{HAZ} c_{HZZ}^{(1)} \bar{e} [g_L^{Z\ell}]^2 g_L^{Zq} Q_q \delta\Gamma \hat{v}^2$ | 0.58 | 0.57 | 0.57 | 0.57 | 0.57 |
| $c_{HZZ}^{(1)} [g_L^{Z\ell}]^2 g_L^{Zq} g_L^{(1),hZq} \delta\Gamma \hat{v}^2$ | -6.9 | -6.5 | -5.7 | -5.5 | -5.4 |

# F   Analysis of up-strange contribution to assess impact of CKM contributions

In the main text we neglect the impact of the CKM matrix in our results by invoking a $U(3)^5$ symmetry. Here we consider the up-strange quark contribution to demonstrate the potential impact of the largest CKM contribution.

Denoting the CKM matrix as $V$, we have for the $V_{12}$ and $V_{21}$ elements [52]:

$$V_{12} = s_{12}c_{13} \sim 0.2136\,, \tag{F.1}$$

$$V_{21} = -\left(s_{12}c_{23} + c_{12}s_{23}s_{13}e^{i\delta}\right) \sim -0.2247 - .0001i\,. \tag{F.2}$$

Notice that the imaginary part of the $V_{21}$ is suppressed by both $s_{23}$ and $s_{13}$ and therefore is negligible.

These values coming from the PDG *assume the SM*. That is, they assume there are no other sources of off-diagonal couplings of the $W$-bosons to quarks or other sources of mixing between the quarks. This is incomplete from the SMEFT perspective, but a full CKM-like analysis is beyond the scope of this work. For a look at the impact of the SMEFT on CKM determinations at dimension-six see [34], for a general look at how SMEFT generated mixing effects can affect the determination of the CKM matrix to all orders in the geoSMEFT see [35]. We will neglect these complications as a first look at the impact of CKM effects on our analysis.

Table 11 gives the integrated cross sections as described Sec. 4.2 and in Tab. 6. In this case, however, we normalize the cross sections to the $ud$ cross sections. This is indicated in the table by the column header $R\frac{(u\bar{s})}{(u\bar{d})}$. The table further shows, in the column labeled CKM, the suppression from assuming SM-like (i.e. $W\bar{\psi}_L\psi_L$) couplings are proportional to the CKM matrix elements above. In the columns labeled 'MFV' further suppression is assumed under the assumption that the only source of $U(3)^5$ breaking is coming from the SM Yukawa couplings. This could be loosened by considering higher dimensional operators that shift the Yukawas [35], but again this requires a full analysis beyond leading order in the SMEFT of the determination of CKM matrix elements. Note that this does not correspond to MFV as defined in [53],[13] but instead is looser and allows for all charged currents arising from $\bar{Q}\gamma_\mu Q$ currents to be proportional to the CKM matrix. This has the effect of suppressing all of the remaining terms by $|V_{ij}|^2$. Finally the last column $R('MFV')/(c\bar{s})$ compares the 'MFV' suppressed result with the $c\bar{s}$ results of Tab. 6. From the last column we see that the the Cabbibo suppressed $u\bar{s}$ terms are almost all of the order 4% of the $c\bar{s}$ contributions, the largest is 40% which is still negligible given the $c\bar{s}$ term is about 10% of the $u\bar{d}$ contribution. The right handed coupling of the $W$ to quarks is forbidden by $U(3)^5$ symmetry even with the looser assumption the only flavor violation comes from the SM Yukawas, so the corresponding element of the table reads "–".

---

[13]In this case all LH currents are suppressed by $m_q$ and are negligible. In our case one could assume that the Wilson Coefficients are given transformation properties under $U(3)^5$ which leave higher dimensional operators invariant under $U(3)^5$, which removes the necessity of this $m_q$ suppression.

Table 11: Reproduction of Table 6 for initial state $\bar{u}s$. The columns labeled $R$ are ratios to the $ud$ contributions. The columns labeled CKM are the $R$ columns scaled by the CKM matrix insertions for SM-like $W$ couplings and the columns labeled 'MFV' use CKM suppression according to the discussion in the text. The last column show the size of the $u\bar{s}$ contributions relative to the $c\bar{s}$ contributions. The right handed coupling of the $W$ to quarks is forbidden by $U(3)^5$ symmetry, even if flavor violation is allowed in the SM Yukawa sector, and so the contribution in this case appears as "–".

| | partons | | | |
|---|:---:|:---:|:---:|:---:|
| $(\mathcal{L}_{\mathrm{eff}}$ dependence$)/N_c$ | $R\frac{(u\bar{s})}{(u\bar{d})}$ | CKM | 'MFV' | $R\frac{\text{'MFV'}}{(c\bar{s})}$ |
| $[c_{HWW}^{(1)}]^2\,\lvert g_L^{W\ell}\rvert^2\,\lvert V_{ij}g_L^{Wq}\rvert^2\cdot 10^{-4}$ | 0.78 | 0.04 | 0.04 | 0.40 |
| $[c_{HWW}^{(2)}]^2\,\lvert g_L^{W\ell}\rvert^2\,\lvert V_{ij}g_L^{Wq}\rvert^2\,\hat{v}^4$ | 0.72 | 0.03 | 0.04 | 0.06 |
| $c_{HWW}^{(1)}c_{HWW}^{(2)}\,\lvert g_L^{W\ell}\rvert^2\,\lvert V_{ij}g_L^{Wq}\rvert^2\,\hat{v}^2$ | 0.78 | 0.04 | 0.04 | 0.04 |
| $c_{HWW}^{(1)}c_{HWW}^{(3)}\,\lvert g_L^{W\ell}\rvert^2\,\lvert V_{ij}g_L^{Wq}\rvert^2\,\hat{v}^2$ | 0.77 | 0.04 | 0.04 | 0.04 |
| $c_{HWW}^{(1)}c_{HWW}^{(4)}\,\lvert g_L^{W\ell}\rvert^2\,\lvert V_{ij}g_L^{Wq}\rvert^2\,\hat{v}^2$ | 0.72 | 0.03 | 0.04 | 0.06 |
| $[c_{HWW}^{(1)}]^2\,\lvert g_L^{W\ell}\rvert^2\,\lvert g_R^{Wq}\rvert^2$ | 0.78 | 0.78 | – | – |
| $\lvert g_L^{W\ell}\rvert^2\,\lvert g_L^{(1),hWq}\rvert^2\,\hat{v}^4$ | 0.58 | 0.58 | 0.03 | 0.10 |
| $c_{HWW}^{(1)}\,\lvert g_L^{W\ell}\rvert^2\mathrm{Re}[(g_L^{(1),hWq})^* V_{ij}g_L^{Wq}]\hat{v}^2$ | 0.73 | 0.16 | 0.03 | 0.04 |
| $c_{HWW}^{(2)}\,\lvert g_L^{W\ell}\rvert^2\mathrm{Re}[(g_L^{(1),hWq})^* V_{ij}g_L^{Wq}]\hat{v}^2$ | 0.71 | 0.15 | 0.03 | 0.04 |
| $c_{HWW}^{(1)}\,\lvert g_L^{W\ell}\rvert^2\mathrm{Re}[(g_L^{(4),hWq})^* V_{ij}g_L^{Wq}]\hat{v}^4$ | 0.57 | 0.12 | 0.03 | 0.10 |
| $c_{HWW}^{(1)}\,\lvert g_L^{W\ell}\rvert^2\mathrm{Re}[(g_L^{(5),hWq})^* V_{ij}g_L^{Wq}]\hat{v}^4$ | 0.56 | 0.12 | 0.03 | 0.12 |
| $c_{HWW}^{(1)}\,\lvert g_L^{W\ell}\rvert^2\mathrm{Im}[(g_L^{(1),hWq})^* V_{ij}g_L^{Wq}]\hat{v}^2\cdot 10^{-3}$ | 0.78 | 0.17 | 0.04 | 0.04 |
| $c_{HWW}^{(1)}\,\lvert g_L^{W\ell}\rvert^2\mathrm{Im}[(g_L^{(4),hWq})^* V_{ij}g_L^{Wq}]\hat{v}^4\cdot 10^{-3}$ | 0.72 | 0.15 | 0.03 | 0.04 |
| $c_{HWW}^{(1)}\,\lvert g_L^{W\ell}\rvert^2\mathrm{Im}[(g_L^{(5),hWq})^* V_{ij}g_L^{Wq}]\hat{v}^4\cdot 10^{-3}$ | 0.68 | 0.15 | 0.03 | 0.05 |
| $c_{HWW}^{(1)}\,\lvert g_L^{W\ell}\rvert^2 g_L^{(2),hWq}V_{ij}g_L^{Wq}\,\hat{v}^4$ | 0.59 | 0.13 | 0.03 | 0.09 |
| $c_{HWW}^{(1)}\,\lvert g_L^{W\ell}\rvert^2 g_L^{(3),hWq}V_{ij}g_L^{Wq}\,\hat{v}^4$ | 0.59 | 0.13 | 0.03 | 0.09 |
| $c_{HWW}^{(1)}\,\lvert g_L^{W\ell}\rvert^2 g_L^{(6),hWq}V_{ij}g_L^{Wq}\,\hat{v}^4$ | 0.56 | 0.12 | 0.03 | 0.12 |
| $c_{HWW}^{(1)}\,\lvert g_L^{W\ell}\rvert^2 g_L^{(7),hWq}V_{ij}g_L^{Wq}\,\hat{v}^4$ | 0.56 | 0.12 | 0.03 | 0.12 |
| $c_{HWW}^{(1)}\,\lvert g_L^{W\ell}\rvert^2 g_L^{(8),hWq}V_{ij}g_L^{Wq}\,\hat{v}^4$ | 0.56 | 0.12 | 0.03 | 0.12 |
| $[c_{HWW}^{(1)}]^2\,\lvert g_L^{W\ell}\rvert^2\,\lvert V_{ij}g_L^{Wq}\rvert^2\delta\Gamma$ | 0.78 | 0.04 | 0.04 | 0.04 |
| $[c_{HWW}^{(1)}]^2\,\lvert g_L^{W\ell}\rvert^2\,\lvert V_{ij}g_L^{Wq}\rvert^2\delta\Gamma^2$ | 0.78 | 0.04 | 0.04 | 0.04 |
| $c_{HWW}^{(1)}c_{HWW}^{(2)}\,\lvert g_L^{W\ell}\rvert^2\,\lvert V_{ij}g_L^{Wq}\rvert^2\delta\Gamma\,\hat{v}^2$ | 0.78 | 0.04 | 0.04 | 0.04 |
| $c_{HWW}^{(1)}\,\lvert g_L^{W\ell}\rvert^2\mathrm{Re}[(g_L^{(1),hWq})^* V_{ij}]g_L^{Wq}\delta\Gamma\,\hat{v}^2$ | 0.73 | 0.04 | 0.04 | 0.05 |

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
