# Peer review of "Higgs associated production with a vector decaying to two fermions in the geoSMEFT"

_SciPost Physics, doi:SciPost Phys. 16, 019 (2024)_

## Round 1 · Referee Report · Anonymous (Referee 1) · 2023-8-22

Strengths
2-The discussion is well framed around the different helicity structures in the processes
3-The results are presented in terms of the couplings of the Lagrangian in the broken phase (eqs 17 through 20), making them flexible with respect to the chosen (geo)SMEFT parametrization
4-EFT effects on W/Z mass and width are included
5-The authors provide their final numerical results in a useful format in ancillary files
Weaknesses
2-Setting the CKM to the unit matrix may be too strong an assumption for the charged current processes
Report
I ask for a few minor clarifications below.
Requested changes
1-Could the authors provide in the text a back-of-the-envelope estimate of the accuracy of their CKM=identity matrix approximation? (For example, roughly how large would a u-sbar initiated contribution to the first row of Table 6 be if Cabibbo mixing were turned on? Are the relative contributions of SM and EFT terms likely to change much relative to the u-dbar initiated case?)
2-Is it understood why $c_{HW}^{(6)}$ gives a much smaller contribution in the neutral current case? If so, it would be helpful to include this in the text.
Some typos:
3-"in associated" -> "in association" in first sentence of abstract
4-unbar one of the fermions in the last line of eq (19)
5-"Mathematic"->"Mathematica" on page 13

---

## Round 1 · Referee Report · Anonymous (Referee 2) · 2023-10-9

Strengths
1- Ready to use result, can be used to provide better constrain versus current results 2-Non trivial calculation 3-Supporting material in form of Mathematica notebook
Weaknesses
2-Sign resolution absence; this is acknowledged in the manuscript.
3-Some expressions can be clarified by better wording.
Report
This manuscript applies a standard approach, geoSMEFT, and include dimension-eight operators to derive the cross section for Higgs associated production with a W± or Z bosons in the Standard Model Effective Field Theory (SMEFT) expansion. These results are valuable for both theorists and experimentalists, since they can readily be applied to obtain experimental bounds on appropriate operators.
In section 2, the authors explains the geoSMEFT tool used throughout the manuscript. Section 3 derive the effective Lagrangian. Section 4 specializes on Higgs' neutral current and charged current channels. This section is helpful in justifying the necessity of including dimension-eight operators.
Overall, this is non-trivial and important result. We would be happy to recommend this manuscript for publication.
Some minor errors the authors may consider fixing: - Section 1, "... the "inverse problem,"..." put comma outside the phrase. - Same sentence, what does "great degeneracy" mean? - In some places, "Wilson Coefficients" should be "Wilson coefficients". - "Mathematic" should be "Mathematica" - Section 4, "... Details ... function integration is discussed ..." should be "are discussed" - Section 5, "...with a W+- or Z bosons...", omit "a".

---

## Round 1 · Referee Report · Anonymous (Referee 3) · 2023-11-3

Strengths
1- Rigorous computation of results 2-Results relevant to experiment 3-Comprehensive
Weaknesses
1- Theoretical criteria seems arbitrary 2-Use of results for more general cases is unclear
Report
The paper presents the inclusive calculation of a Higgs produced in association with vector bosons for a selected array of dimension 8 operators. Results are relevant for experimental searches but the paper could be improved in a number of ways
1- The authors say, "That the geoSMEFT classifies all three point functions is a basis dependent statement". This is unclear to me, is there a different geoSMEFT for every different basis?
2- The programme is spelled out in 2.2. In essence, as I understand it, the geoSMEFT is used to select or discard dimension 8 operators. However what makes geoSMEFT a privileged set up to have such a say; does it offer for example any improvement on the convergence of the 1/\Lambda expansion? Does it match more naturally to a type of models?
3-In addition to a number of reasons, the authors say that the opertor in eq 12 can be discarded because it leads to ghosts. How can a perturbative treatment of an EFT, specifically designed to include only certain degrees of freedom, have its spectrum modified by the inclusion of an operator?
4-Results are given in tables such as tab~4. Can these results be combined if one wants to include both dimension 6 linear and squared contributions and dimension 8 or would these require new computations?
Requested changes
Address the points in the report

---

## Round 2 · Author Response

Dear Referees, We apologize for the delay in response, the referee reports were received over the course of months and we wished to delay response until the final one was receive and the editor had made their first recommendation. We hope we have addressed your concerns.

Response to Referee 1: We thank the referee for their constructive comments and have attempted to respond to all concerns. In particular their primary concern about the up-strange contributions: 1) We have added Appendix F where we reproduce Table 6 for the u \bar s contribution. The appendix is referenced in the text . We compare the size of the integrations with the u \bar d case, this multiplied by the relevant insertion of VCKM for SM-like W couplings, under an 'MFV'-like assumption made in the main text, as well as comparing the MFV-like case to the contribution from c \bar s. In doing so we find that under the MFV-like assumption these terms are negligible compared with the c \bar s contribution (order 4-40% of the c \bar s contributions which are already small compared to u \bar d). We stress again in the appendix that a proper consideration of off diagonal couplings of the Ws requires a full dimension-eight analysis of CKM data and a proper definition of the CKM matrix in the presence of all these couplings.

2) Could the referee clarify in what sense cHW6 gives a smaller contribution? In the ancillary files, the contribution to the NC process (normalized to the SM, alpha scheme, with the WCs in units of the vev) is weighted by 15.7 while in the CC case it is weighted by 14.1.

3) We have corrected this mistake.

4) We have corrected this mistake.

5) We have corrected this mistake.

Ref 2: We thank the referee for their careful reading of the article, we have made the following changes corresponding to the minor comments and numbered according to their order in the report: 1) We have retained that the commas should be placed inside the quotes as this is our understanding of punctuation placement in quotes regardless of whether they are a part of the quote. 2) We changed the sentence to drop the phrase "great degeneracy" and instead more carefully explain what we mean: "The ``inverse problem,'' or that there are a large number of UV completions of the SM which imprint on the same subset of operators at dimension six." 3) We searched the document for "Coefficients" with capital "C" and replaced the instances found with "coefficients" 4) We found and replaced one instance of "Mathematic" with "Mathematica" 5) We changed the language to "Details of the phase space and parton distribution function integrations are discussed" 6) We opted for keeping "a" but changed "bosons" to "boson" as only one W or Z is present in the given processes.

Ref 3 We thank the referee for their comments, particularly on the language surrounding the geoSMEFT. We understand the geoSMEFT requires careful explanation and are trying to improve with each iteration. 1) we have added a paragraph just before the last paragraph of the intro to section 2 stating: The “geoSMEFT” as used in this article and the literature corresponds to the choice of fully classifying all two- and three-point functions. This choice greatly simplifies calculations involving resonant physics. One could, however, apply the geometric methodology to any basis of SMEFT operators by defining a different minimal set of field-space connections.

2) We amended point 2 of the procedure for making the dimension-eight operators consistent with the geoSMEFT to state:

"2. Removing these operators from the dimension-eight operator basis via integration by parts and the equations of motion. That is, the operators’ effects are moved to other operators consistent with the geoSMEFT."

The second sentence was added to clarify we aren't discarding operators, we are simply moving their effects to operators which are consistent with the geoSMEFT basis but not included in one or both of the bases of dimension-eight operators we cite. The advantage offered by the geoSMEFT is in simplifying calculations (especially in regards to input parameter schemes), it has not been shown to improve convergence or to match more naturally to UV completions.

3) The modification of the spectrum occurs above the cutoff of the theory and therefore does not present a problem in the EFT. In the reference article, arXiv:1405.5412, in Eq.2.4 this can be seen to correspond to 1/c, with c the Wilson coefficient of the operator of the dimension-six analogue of our Eq.12. This corresponds to Lambda^2 and so we can see the connection between the UV and IR.

This can also be inferred from, for example, arXiv:2007.00565 Eq. 6.6 in the last line. Here the Lee-Wick ghost's mass should actually correspond to the resonance of the "K_\mu" particle which has been integrated out.

We have not added this discussion to the article as we feel it goes too far astray of the discussion and the citation suffices. We chose to include the citation in our discussion in the first place as this category of operators is preferred in the dimension-eight basis of operators we cite, arXiv:2005.00008.

4) The tables show the contribution from a given term using the effective Lagrangians such as that in Eq.18, as such one could obtain the dimension-six squared contributions absent the dimension-eight operator contributions by carefully reconstructing the appropriate terms from the tables and Appendices A and B. This could also be implemented through the ancillary files. To clarify this, focusing on the ancillary files which are the simplest way to approach this, in the paragraph below Eq.24 we have added the following statement: "The ancillary files are designed for calculations at fixed order in 1/Λ^2, however with minor alterations an interested reader could modify them to produce only the dimension-six squared contributions at order 1/Λ^4."

---

## Round 2 · List of Changes

Please see Author comments where we have line-by-line responded to the referee's requests for corrections

---

## Editorial Decision

published